# Scientific publications that use promotional language in the abstract receive more citations and public attention
Olga Stavrova [1] ✉, Bennett Kleinberg[2], Anthony M. Evans[3] & Milena Ivanović[4]

Researchers often use promotional language ("hyping") in scientific publications to draw attention to their findings. Here we examined whether promotional language is indeed associated with higher academic impact and public attention. A content analysis of over 130,000 abstracts published in three major interdisciplinary outlets (PNAS: 84,603; Science: 25,142; Nature: 26,870) between 1991 and 2023 showed that promotional language predicted more citations and more full-length paper views, more paper mentions in online media and higher Altmetric scores. Further, additional analyses by first and last author gender (first female author $n = 15,368$, first male author $n = 32,873$, last female author $n = 10,218$, last male author $n = 46,606$) showed that despite women being often advised to engage in more self-promotion, following this advice was not significantly associated with a smaller gender gap in impact indicators. If anything, promotional language predicted a larger gender gap with men (vs. women) receiving even more citations, paper views and mentions in the media. Our findings highlight the role of communication strategies in academic impact and public attention, as well as gender diversity in academia.

Novelty is important for impactful science. Scientists often use promotional language ("hyping") to emphasize the novelty and importance of their work[1]. The use of promotional language in published research papers has soared in the last decades[1–3]. This trend is sometimes seen as problematic, as "hyping" might undermine perceptions of scientific rigor and reproducibility, tarnish author reputations, and even erode the collective trust in science and endanger the public[2,4–7]. Given these potential risks, it's important to understand how the use of promotional language affects scientific and public engagement with research. The present study sought to answer this question. We examined the associations between the use of promotional language in over 130,000 scientific papers' abstracts in three major academic outlets – Science, Nature, and the Proceedings of the National Academy of Science (PNAS) and these papers' academic impact (citation and view count) and the public attention (Altmetric score (www.altmetric.com), number of mentions on Twitter/X and in online media in general). Further, drawing from the literature on gender differences in the advantages of self-promotion[8,9], we examined whether the use of promotional language is associated with equal advantages for male and female scientists.

## Promotional language in science communication

In recent decades, scholars have observed a shift in the style of scientific writing, believed to mirror broader trends in academic publishing and the increasing competition for attention and recognition[10]. Analyses of research articles across various fields reveal a decline in cautious / uncertain language and the growing use of promotional language. For example, Yao et al.[10] observed a decrease in the use of hedges—linguistic markers of uncertainty or caution—in papers published in *Science* over the past 25 years. A similar shift was documented by Poole et al. in biochemical research articles, with epistemic stance markers expressing certainty (boosters) increasing, while those indicating doubt or uncertainty (hedges) decreasing[11]. Scholar have observed an increase in the use of promotional language, also referred to as hyping, as well. Promotional language describes the terms (of both positive and negative valence) designed to promote, exaggerate, or embellish research findings (e.g., essential, ultimate, unprecedented, invaluable, promising, tremendous, alarming). Specifically, Hyland and Jiang found a significant rise in the use of such terms in articles from four academic disciplines between 1965 and 2015[2]. It has been suggested that by reducing hedging and embracing promotional language, authors aim to present their

[1]University of Lübeck (Germany) and Tilburg University (Netherlands), Lübeck, Germany. [2]Tilburg University (Netherlands) and University College London, London, UK. [3]Allstate Corporation, Northfield, USA. [4]Ipsos Strategic Marketing, Belgrade, Serbia. ✉e-mail: olga.stavrova@uni-luebeck.de

findings more assertively, enhancing their appeal to editors, reviewers, and readers[2,12].

Does the adoption of promotional language achieve its goal of enhancing a publication's academic impact and persuading the public in its significance? Several theoretical accounts support this possibility. While highlighting the significance of the work, promotional language almost inevitably downplays its limitations and minimizes any potential uncertainty surrounding its findings. Humans generally find uncertainty unpleasant; it is associated with heightened stress responses to negative stimuli and risk information[13,14]. This aversion to uncertainty also extends to perceptions of experts. Individuals tend to distrust doubtful, uncertain advisors [15,16] as uncertainty is frequently interpreted as a lack of competence or expertise[17]. Further, people are less likely to recommend newspaper articles that use more uncertain language and to share corporate advertisements that include more linguistic expressions of uncertainty (e.g., hedging)[18,19]. Most importantly, recent studies have shown that incorporating uncertain language in science communication reduces trust in the message content as well as in its source, and even hinders information diffusion[20–22].

On the opposite, the use of promotional language has been associated with positive outcomes. Recent analyses of grant applications connected the use of promotional words in grant proposals with increased funding rates [23–25]. For example, Qiu et al. analyzed over 11,500 biomedical grant applications and found that the use of promotional language in grant proposals was positively associated with funding success[25]. Further, Peng et al. reported that grant proposals with more promotional language were more likely to result in publications in high impact journals[24]. Not surprisingly, the use of promotional language in funded NIH grant applications has risen from 1985 to 2020[1]. While grants are central for innovation and scientists' careers [26,27], grant proposals, for the largest part, outline future research plans and therefore cannot fully capture the academic and public impact of science. In contrast to grant proposals, "hyping" in academic publications might be especially consequential, as published work receives much larger readership, underpins future research, and shapes policy and public opinion.

Two recent papers have linked the use of "positive words" (e.g., favorable, promising, excellent, supportive, unique) in academic titles and abstracts with a higher citation count, with an exception of papers published in "low impact" clinical journals (defined by the authors as having an impact factor below ten)[28,29]. While these findings suggest that promotional language might boost impact, promotional language should not be equated with positive language. Promotional language includes negatively connoted words as well (e.g., dire, dismal, daunting, alarming), which can be used to signal the significance of the work (e.g., "Alarming trends in US domestic violence during the COVID-19 pandemic"[30]). Therefore, it remains unclear whether publications that use more promotional language achieve higher impact.

The use of promotional language might be particularly consequential in abstracts. Abstracts are the first touchpoint with an academic output: while the full text of a publication is typically read only by a specialized audience, abstracts are accessed by a broader academic readership. Further, abstracts are often criticized for overstating significant findings compared to the respective papers' full texts and are therefore even referred to as a "promotional genre"[31,32]. For example, Shinohara et al. analysis of psychiatry trial papers revealed that 30% of abstracts contained exaggerated claims relative to the full text[32]. In the present research, we examined the associations between the use of promotional language in over 130,000 scientific papers' abstracts in three major academic outlets – Science, Nature, and the Proceedings of the National Academy of Science (PNAS) – published between 1991 and 2023, and these papers' academic impact and public attention.

## Self-promotion, gender and the backlash hypothesis

Even though promotional language might drive research impact, it may at the same time contribute to impact disparities among genders. As prior research suggests that women are, on average, less likely to use promotional language[25,28], this rhetorical strategy could represent an additional barrier to

recognition – thereby exacerbating gender-based inequalities in academic impact. Female academics continue to lag behind in terms of key output criteria, such as citations and visibility[33–37]. For example, an analysis of citation numbers of articles from five high-impact medical journals found that articles authored by women as both primary and senior authors received approximately half as many citations as those authored by men[33]. Women's lower tendency to engage in self-promotion has been blamed for their underrepresentation in science[38]. For example, female academics tend to use more modest language in presenting their work in paper abstracts compared to men[25,28], engage less in self-citation behavior[39] and are less likely to promote their papers on social media[40].

Not surprisingly, women are recommended to self-promote more if they want to get ahead[41]. Yet, a theoretical account – the backlash hypothesis – suggests that self-promotion might backfire for women. Studies have shown that women who do engage in self-promotion tend to experience a backlash, as self-promotion is incongruent with the gender stereotype of modesty[9,42,43]. For example, women who exhibit self-promoting behaviors tend to be perceived as less likeable, which obstructs their careers[9,44]. For example, female job candidates who engage in self-promotion during the job interview receive lower interview evaluations and face reduced hiring likelihood[45,46]. In another study, even though using assertive language in written communication positively affected the credibility of both female and male leaders, female leaders were less likely to use it, undermining their influence and impact[47]. Finally, one recent study has compared the consequences of self-promotion on social media for female and male academics showing that Twitter posts written by female (vs. male) scientsts receive less user engagement[40].

Yet, it remains to be explored whether the use of promotional language in academic papers is differently associated with academic impact and public attention depending on scientist gender. Here, we put the backlash idea to a field test by using objective measures of both self-promotion (share of promotional language in paper abstracts) and academic achievement (papers' academic impact and public attention) across over 50,000 publications by male and female scientists. We focused on first and last authors' gender, as these often represent the positions associated with the strongest contributions.

## Methods
### Dataset
We downloaded all the records that were accessible on the Web of Science platform on June 5, 2023, for three publications: Proceedings of the National Academy of Sciences (PNAS), Science and Nature. We did not apply any filters. After obtaining the data, we removed duplicate records (records with the same DOI appearing in the dataset more than once), records without a DOI or abstract, and records that did not include the publication of primary research findings (e.g., letter, correction notes, editorial materials). We retained the entries of the type "Article" ($n = 135,657$), "Article; Proceedings Paper" ($n = 779$), "Article; Publication with Expression of Concern" ($n = 11$), and "Article; Retracted Publication" ($n = 168$). We decided to keep retracted publications and publications with an expression of concern as they usually continue to receive citations[48]. The final dataset included 136,615 abstracts of research papers published between 1991 and 2023 (PNAS: 84,603; Science: 25,142; Nature: 26,870).

The study was not pre-registered. Data and analyses scripts can be accessed at: https://osf.io/z3x4c/.

### Measures
*Promotional language.* We used the dictionary of "promotional language" that included a list of 139 promotional words (e.g., unprecedented, enormous, incredible, devastating, alarming). The dictionary has been extensively validated in prior work[1,23,24]. The complete list of promotional words in presented in the Supplementary Information File. For each abstract, we computed the share of promotional words (median = 0.73 out of 100 words, $M = 0.86$, $SD = 0.83$, range $0-11.11$). An average abstract contained 0.86% promotional words (i.e., 1.5 words given the average abstract length of 180 words).

*Impact*. We measured papers' academic impact and public attention using six metrics. Three were obtained from the Web of Science database: citation count (per year), number of times the record has been accessed on the Web of Science platform since last 180 days and since 2013 (per year). Citation count refers to the number of citations the record received in journals, conference proceedings, book series, and other papers that are indexed in the Web of Science. Given that the citation count likely increases with the time the paper has been available, we used the citation count per year for our analyses. Usage count since last 180 days and since 2013 reflect the number of times the record has been accessed on the Web of Science platform (e.g. by clicking on the link to the publisher's website or by saving it in a bibliographic management tool). For usage count since last 180 days, we took its raw value; for usage count since 2013, we computed the count per year (again, correcting for the fact that usage numbers accumulate with the time the paper is available online).

The other three outcomes reflected the public attention and were obtained from Altmetric, a platform that tracks media mentions of research papers, using its API. The public attention metrics were available for $n = 64,693$ records. They included the Altmetric score (a measure of the record's mentions outsides of the academic circles, e.g., in the press, policy documents or Wikipedia[49]); the number of online mentions, referred to as "posts", which reflects the number of all mentions in all types of online documents tracked by Altmetric (Facebook, LinkedIn, blogs, reddit, news etc.); and number of mentions on Twitter / X. We specifically included the count of Twitter mentions as it has been one of the most popular online social network for science communication at the time of data collection. The distribution of all measures is shown in Supplementary Fig. 1 in the Supplementary Information File.

*Author gender*. We used gender_guesser, an AI algorithm[50], to identify the author gender based on their first name. We focused on first (primary or leading) and last (senior) authors' gender, as these usually represent the strongest contributions – a practice common in prior studies[28,33]. The algorithm is based on over 45,000 names and provides the following categories: male name, female name, mostly male name, mostly female name, androgynous (same probability of male and female), unknown (name is unknown). Among fist authors, 15,368 records were categorized as having a female first author, 32,873 were categorized as having a male first author (of the remaining records, 1138 were "mostly female", 1609 were "mostly male", 5893 were androgynous and the rest were classified as "unknown", as the respective records only included the first name's first letter (e.g., M. Smith)). Among last authors, 10,218 records were categorized as having a female last author, 46,606 were categorized as having a male last author (of the remaining records, 806 were "mostly female", 1513 were "mostly male", 4047 were androgynous and the rest were unknown). Our analyses used the subsample for which author gender could be unequivocally classified as male or female (first author gender: $n = 48,241$; last author gender: $n = 56,824$). To ensure the validity of the automatic name classification, we randomly selected 100 records that were classified as having a male first author and 100 records that were classified as having a female first author for a manual classification. The manual classification was done by the first author who could classify 162 (out of 200) names but lacked the cultural knowledge to classify the remaining 38 records that were therefore submitted to genderize.io classification tool[51]. The automatic classifier used here (gender_guesser) and the human rater / genderize.io agreed in 197 cases out of 200. The three cases where disagreement emerged were resolved by finding the respective researcher's profile on the web: in two cases, the human rater / genderize.io provided the correct classification and in one case, gender_guesser was correct. Overall, this low error rate (1.5%) suggests that the automatic classification provides reliable information on author gender.

*Control variables*. We additionally obtained the information on records' features that have been previously associated with higher impact and / or promotional language in prior studies[2,24,52,53]: journal (Science, Nature, PNAS), publication year, word count (number of words in the abstract), number of authors, institutional diversity (number of listed affiliations divided by the number of authors; higher score indicates more collaborations across different institutions), overall semantic positivity (summary variable Tone that quantifies the overall semantic positivity of the text, obtained using the Linguistic Inquiry and Word Count software[54]), and the indicator of a subject field. The subject field indicator was obtained by applying an algorithm trained to classify research papers' abstracts into one of 20 fields of study[55]. We used a medium Multi-Layer Perceptron model as it showed good recall (.82) and precision (.87) values. The analyses were done using the Field of Study Classification python module. Some of the 20 fields included a very small number of papers: for example, there was only one paper in the field of philosophy and religious sciences, 14 in the field of legal studies and 15 in the field of creative arts and writing sciences (to compare, the most well-represented field – biological sciences – included almost 65,000 records). Therefore, we recoded the 20 fields of study into three broader academic areas: formal and natural sciences ($n = 105,177$), health and medicine ($n = 27,302$), and social sciences and humanities ($n = 4136$).

## Analytic strategy

The effect of promotional language on impact indicators was estimated using regression models. As citation count, usage count, count of mentions in tweets and online posts are over-dispersed count variables with a large number of zeros (for variable distributions, see Supplementary Fig. 1), we used negative binomial regression in the respective analyses. The Altmetric score does not represent a count, yet its distribution was highly skewed (skew = 13.82, kurtosis = 348.09). To correct for its skewness, we computed its natural logarithm (skew = 0, kurtosis = -0.44). For the analyses of the Altmetric score, we used a linear regression on the log-transformed Altmetric score.

For each outcome, we tested the effect of promotional language first in a model without covariates and then in a model with the control variables listed above. We also conducted a series of robustness checks. First, as Altmetric started collecting data in 2011, we repeated the analyses of public attention using only the data on the papers published after 2010. Second, as the data collection took place in June 2023, the impact metrics for papers published in 2023 may be unreliable. Therefore, another robustness check was conducted by repeating the analyses while excluding all papers from 2023. Third, as some of the "promotional language" terms might have several meanings (e.g., "stellar" as "exceptional" vs. describing stars in astronomy) and can thus artificially inflate the promotional language score for some papers, we repeated the analyses while excluding abstracts with an exceptionally high (3 *SD* above the mean) share of promotional words. The results of all robustness checks were consistent with the main analysis and are reported in the Supplementary Information File (Supplementary Tables 1-9). The analyses were not pre-registered.

## Results
### Correlations among impact and public attention indicators

All indicators of academic impact were positively associated with indicators of public attention, ranging from $r = 0.10$, 95%CI [0.09, 0.11], $p < 0.001$ (between views since 2013 and mentions in tweets) and $r = 0.25$, 95%CI [0.24, 0.26], $p < 0.001$ (between citation count and Altmetric score). The correlation matrix is shown in Supplementary Table 11.

### Impact

Negative binomial (in case of count outcomes) and linear (in case of Altmetric score) regression analyses showed that promotional language was significantly associated with impact indicators in all models. The model coefficients are shown in Table 1. We estimated the effect of promotional language with and without adjusting for a number of covariates, including journal, publication year, abstract length, subject field, number of authors and affiliations, and overall abstract semantic positivity (see Methods for measurement details). Depending on the model specification, increasing promotional language by 1% (i.e., adding about 2 promotional words) predicted 9−14% more citations per year (model without controls: IRR =

# Table 1 | Promotional language, academic impact and public attention

| Predictors | Citations IRR | CI | p | Views 180 IRR | CI | p | Views 2013 IRR | CI | p | Altmetric b | CI | p | Tweets IRR | CI | p | Posts IRR | CI | p |
|---|---|---|---|---|---|---|---|---|---|---|---|---|---|---|---|---|---|---|
| (Intercept) | 13.25 | 13.14 – 13.37 | <0.001 | 2.84 | 2.80 – 2.89 | <0.001 | 5.23 | 5.17 – 5.29 | <0.001 | 0.44 | 0.42 – 0.45 | <0.001 | 65.73 | 64.44 – 67.06 | <0.001 | 98.56 | 96.80 – 100.36 | <0.001 |
| Promotional language | 1.14 | 1.13 – 1.15 | <0.001 | 1.37 | 1.35 – 1.39 | <0.001 | 1.37 | 1.35 – 1.38 | <0.001 | 0.08 | 0.07 – 0.09 | <0.001 | 1.12 | 1.10 – 1.14 | <0.001 | 1.11 | 1.09 – 1.12 | <0.001 |
| Observations | 136615 | | | 136615 | | | 136615 | | | 64693 | | | 64693 | | | 64693 | | |
| $R^2$ | 0.015 | | | 0.027 | | | 0.036 | | | 0.003/0.003 | | | 0.005 | | | 0.005 | | |
| (Intercept) | 5.86 | 5.63 – 6.09 | <0.001 | 0.20 | 0.18 – 0.21 | <0.001 | 0.16 | 0.15 – 0.16 | <0.001 | 0.42 | 0.38 – 0.46 | <0.001 | 2.26 | 2.08 – 2.45 | <0.001 | 13.94 | 12.89 – 15.08 | <0.001 |
| Promotional language | 1.09 | 1.08 – 1.10 | <0.001 | 1.16 | 1.14 – 1.17 | <0.001 | 1.14 | 1.13 – 1.15 | <0.001 | 0.03 | 0.03 – 0.04 | <0.001 | 1.04 | 1.03 – 1.05 | <0.001 | 1.05 | 1.03 – 1.06 | <0.001 |
| Journal [nature] | 2.95 | 2.91 – 2.99 | <0.001 | 4.00 | 3.91 – 4.10 | <0.001 | 4.48 | 4.41 – 4.56 | <0.001 | 0.77 | 0.76 – 0.78 | <0.001 | 3.32 | 3.23 – 3.42 | <0.001 | 3.21 | 3.13 – 3.30 | <0.001 |
| Journal [science] | 3.00 | 2.96 – 3.05 | <0.001 | 4.56 | 4.43 – 4.70 | <0.001 | 5.32 | 5.21 – 5.43 | <0.001 | 0.82 | 0.80 – 0.84 | <0.001 | 2.98 | 2.88 – 3.09 | <0.001 | 3.26 | 3.15 – 3.37 | <0.001 |
| Year | 1.01 | 1.01 – 1.01 | <0.001 | 1.10 | 1.10 – 1.10 | <0.001 | 1.12 | 1.12 – 1.12 | <0.001 | 0.04 | 0.04 – 0.04 | <0.001 | 1.15 | 1.15 – 1.15 | <0.001 | 1.08 | 1.07 – 1.08 | <0.001 |
| Word count | 1.00 | 1.00 – 1.00 | <0.001 | 1.00 | 1.00 – 1.00 | 0.447 | 1.00 | 1.00 – 1.00 | <0.001 | 0.00 | 0.00 – 0.00 | <0.001 | 1.00 | 1.00 – 1.00 | <0.001 | 1.00 | 1.00 – 1.00 | <0.001 |
| Field [formal & natural] | 0.88 | 0.85 – 0.91 | <0.001 | 0.99 | 0.94 – 1.05 | 0.748 | 1.09 | 1.05 – 1.13 | <0.001 | -0.55 | -0.58 – -0.53 | <0.001 | 0.26 | 0.25 – 0.28 | <0.001 | 0.32 | 0.30 – 0.33 | <0.001 |
| Field [medical & health] | 1.08 | 1.04 – 1.11 | <0.001 | 0.57 | 0.54 – 0.61 | <0.001 | 0.69 | 0.66 – 0.72 | <0.001 | -0.44 | -0.46 – -0.41 | <0.001 | 0.44 | 0.42 – 0.46 | <0.001 | 0.53 | 0.50 – 0.56 | <0.001 |
| Author number | 1.01 | 1.01 – 1.01 | <0.001 | 1.00 | 1.00 – 1.00 | <0.001 | 1.00 | 1.00 – 1.00 | <0.001 | 0.00 | 0.00 – 0.00 | <0.001 | 1.01 | 1.01 – 1.01 | <0.001 | 1.01 | 1.01 – 1.01 | <0.001 |
| Institutional diversity | 0.92 | 0.91 – 0.93 | <0.001 | 0.99 | 0.97 – 1.01 | 0.357 | 1.00 | 0.98 – 1.01 | 0.518 | 0.07 | 0.06 – 0.08 | <0.001 | 1.32 | 1.29 – 1.35 | <0.001 | 1.22 | 1.19 – 1.25 | <0.001 |
| Positivity | 1.00 | 1.00 – 1.00 | <0.001 | 1.01 | 1.01 – 1.01 | <0.001 | 1.01 | 1.01 – 1.01 | <0.001 | 0.00 | 0.00 – 0.00 | <0.001 | 1.00 | 1.00 – 1.00 | 0.506 | 1.00 | 1.00 – 1.00 | 0.496 |
| Observations | 136612 | | | 136612 | | | 136612 | | | 64690 | | | 64690 | | | 64690 | | |
| $R^2$ | 0.443 | | | 0.503 | | | 0.782 | | | 0.294 / 0.294 | | | 0.565 | | | 0.437 | | |

Note. $b$ = unstandardized regression coefficient; IRR = Incidence Rate Ratio; CI = 95% Confidence Interval. Citations: citation count per year; Views 180: the number of times the record has been accessed on the Web of Science platform (e.g. by clicking on the link to the publisher's website or by saving it in a bibliographic management tool) in the last 180 days; Views 2013: the number of times the record has been accessed on the Web of Science platform (e.g. by clicking on the link to the publisher's website or by saving it in a bibliographic management tool) since 2013, per year; Altmetric: Altmetric score; Tweets: number of mentions on Twitter/X; Posts: overall number of mentions in any type of online document (news, social media, policy documents etc.). Promotional language: share of promotional language in the abstract; Journal: PNAS is reference category; Year: publication year; Word count: number of words in the abstract; Field: social and humanities is reference category; author number: number of authors; institutional diversity: number of listed affiliations divided by the number of authors, higher score indicates more collaborations across different institutions; Positivity: overall semantic positivity of the abstract, obtained using the Linguistic Inquiry and Word Count software (LIWC, version 2022). $R^2$: for negative binomial models, Nagelkerke's $R^2$ is reported, for a linear model (Altmetric sore), multiple $R^2$/adjusted $R^2$ are reported.

1.14, 95%CI [1.13–1.15], $p < 0.001$; model with controls: IRR = 1.09, 95%CI [1.08-1.10], $p < 0.001$), 14–37% more paper access within the last 180 days (model without controls: IRR = 1.37, 95%CI [1.35–1.39], $p < 0.001$; model with controls: IRR = 1.16, 95%CI [1.14-1.17], $p < 0.001$) as well as each year since 2013 (model without controls: IRR = 1.37, 95%CI [1.35–1.38], $p < 0.001$; model with controls: IRR = 1.14, 95%CI [1.13-1.15], $p < 0.001$), a 3−6% higher Altmetric score (model without controls: $b = 0.08$, 95%CI [0.07-0.09], $p < 0.001$; model with controls: IRR = 0.03, 95%CI [0.03-0.04], $p < 0.001$), 4−12% more Twitter mentions (model without controls: IRR = 1.12, 95%CI [1.10–1.14], $p < 0.001$; model with controls: IRR = 1.04, 95%CI [1.03–1.5], $p < 0.001$) and 5−11% more online mentions overall (model without controls: IRR = 1.11, 95%CI [1.09-1.12], $p < 0.001$; model with controls: IRR = 1.05, 95%CI [1.03–1.06], $p < 0.001$), see Fig. 1. The results held when restricting the sample to the papers published after Altmetric was founded in 2011, excluding papers published in 2023 and excluding outliers (see robustness check in the Supplementary Tables 1–9).

### Author gender

Differences in impact and promotional language between papers with a female (vs. male) lead and senior authors are summarized in Table 2. Abstracts of papers with female (vs. male) *first* author included significantly (6%) less promotional language ($t$ (31632) = 6.77, $p < 0.001$, Cohen's $d = 0.06$, 95%CI [0.05, 0.08]); in contrast, abstracts of papers with a female *last* author did not significantly differ in the share of promotional words compared to abstracts of papers with a male *last* author ($t$ (15380) = -0.22, $p = 0.828$, Cohen's $d = 0$, 95%CI [-0.02, 0.02]). On average, papers with a female (vs. male) first or last author had a significantly poorer academic impact (percentage difference ranged from 18% to 34%) (first author, citations: $t$ (42308) = 11.64, $p < 0.001$, Cohen's $d = 0.10$, 95%CI [0.08, 0.12], views in 180 days: $t$ (42714) = 12.33, $p < 0.001$, Cohen's $d = 0.10$, 95%CI [0.09, 0.12]; views since 2013: $t$ (43349) = 15.34, $p < 0.001$, Cohen's $d = 0.13$, 95%CI [0.11, 0.15]; last author, citations: $t$ (19179) = 9.36, $p < 0.001$, Cohen's $d = 0.09$, 95%CI [0.06, 0.11], views in 180 days: $t$ (16573) = 4.69, $p < 0.001$, Cohen's $d = 0.05$, 95%CI [0.03, 0.07]; views since 2013: $t$ (18032) = 7.39, $p < 0.001$, Cohen's $d = 0.07$, 95%CI [0.05, 0.09]), and papers with a female (vs. male) first (but not last) author had a significantly lower Altmetric score and significantly less online mentions overall (percentage difference ranged from 5% to 13%) (first author, Altmetric: $t$ (20898) = 7.92, $p < .001$, Cohen's $d = 0.09$, 95%CI [0.07, 0.11]; online mentions: $t$ (18892) = 2.07, $p = 0.039$, Cohen's $d = 0.03$, 95%CI [0, 0.05]; last author, Altmetric: $t$ (11261) = 0.64, $p = 0.122$, Cohen's $d = -0.01$, 95%CI [-0.03, 0.02]; online mentions: $t$ (13455) = 0.86, $p = 0.388$, Cohen's $d = -0.01$, 95%CI [-0.03, 0.02]).

Did women who used more promotional language in their paper abstracts show a smaller impact gap with men? To answer this question, we regressed each outcome on promotional language, author gender and their interaction. We computed different models using first and last author gender. We tested 12 models: two models – with and without covariates – per outcome. The model coefficients are shown in Table 3 (first author gender) and Table 4 (last author gender). The *first* author gender by promotional language interaction was significant in only 3 out of 12 models (citations, model with controls: IRR = 0.98, 95%CI [0.96, 1.00], $p = 0.032$, mentions in tweets, model with controls: IRR = 0.94, 95%CI [0.91-0.98], $p = 0.002$; overall online mentions, model with controls: IRR = 0.95, 95%CI [0.92-0.99], $p = 0.008$). When significant, the interaction suggested that the impact gap between female- and male-led papers increased with increasing promotional language, see Fig. 2. The *last* author gender by promotional language interaction reached significance in 7 out of 12 models, see Fig. 3 (citations, model without controls: IRR = 0.95, 95%CI [0.92, 0.98], $p < 0.001$, model with controls: IRR = 0.97, 95%CI [0.94, 0.99], $p = 0.004$; views since 180 days, model without controls: IRR = 0.93, 95%CI [0.88, 0.97], $p = 0.002$; views since 2013, model without controls: IRR = 0.96, 95%CI [0.93, 0.99], $p = 0.015$; Altmetric, model without controls: $b = -0.04$, 95%CI [-0.06, -0.01], $p = 0.003$, model with controls: $b = -0.03$, 95%CI [-0.05, 0], $p = 0.015$; online mentions, model without controls: IRR = 0.95, 95%CI [0.91, 1,00], $p = 0.045$). Again, the impact gap between female- and male-last author

papers increased with increasing promotional language, Fig. 3. The planned robustness checks led to similar conclusions (see Supplementary Tables 7-9).

### Exploratory analyses: Temporal evolution

Recent research has shown that the positive association between positive words and citation count has been decreasing over time, a finding potentially explained by the readership getting accustomed to positive language, reducing its effectiveness as a signal of novelty[29]. We explored to what extent a similar trend exists with respect to promotional language. We repeated the analyses with impact indicators as dependent variables, while using promotional language, publication year (entered as a continuous predictor) and the interaction between the two as predictors. The interaction term was significant with respect to all impact indicators and persisted when including the control variables (same as in the main analyses), with the exception of views since 180 days: citations, model without controls: IRR = 0.996, 95%CI [0.995, 0.997], $p < 0.001$, model with controls: IRR = 0.996, 95%CI [0.995, 0.997], $p < 0.001$; views since 180 days, model without controls: IRR = 0.997, 95%CI [0.996, 0.999], $p < 0.001$, model with controls: IRR = 0.999, 95%CI [0.998, 1.001], $p = 0.251$; views since 2013, model without controls: IRR = 0.994, 95%CI [0.993, 0.995], $p < 0.001$, model with controls: IRR = 0.996, 95%CI [0.995, 0.997], $p < 0.001$; Altmetric, model without controls: $b = 0.002$, 95%CI [0.001, 0.003], $p = 0.004$, model with controls: $b = 0.001$, 95%CI [0.001, 0.002], $p = 0.002$; mentions in tweets, model without controls: IRR = 1.005, 95%CI [1.003, 1.007], $p < 0.001$, model with controls: IRR = 1.005, 95%CI [1.003, 1.007], $p < 0.001$; overall online mentions, model without controls: IRR = 1.005, 95%CI [1.004, 1.007], $p < 0.001$, model with controls: IRR = 1.006, 95%CI [1.004, 1.008], $p < 0.001$. The pattern of the interaction depended on the specific impact indicator used. Specifically, we found that the association between promotional language and impact has diminished over time when impact is measured as citation count (per year). In contrast, with respect to other five indicators of impact (usage per year since 2013, usage since the last 180 days, Altmetric score, number of mentions in tweets and online posts), the positive association between promotional language use and impact increased with time. The model coefficients are presented in Supplementary Table 10 and Supplementary Fig. 2. We return to this finding in the discussion.

### Discussion

Researchers increasingly rely on promotional language to entice academic audiences and the public[1,2]. But are papers that use more promotional language indeed more impactful? Across over 130,000 abstracts published in three top academic journals in the last thirty years, we observed that papers whose abstracts contained more promotional words were accessed more often, received more citations and more online mentions, including on the social media. These findings extend the recent work showing that grants using more promotional language are more likely to get funded[24,25]. It's important to note that the link between promotional language and academic impact as well as public attention was somewhat smaller than in the context of grant funding, but yet substantial: e.g. increasing promotional language by just 1% (i.e., adding about 2 promotional words, given an average abstract length of 180 words) was associated with up to 14% more citations per year. For a paper with a citation count per year of 15 (sample mean) that has been available for 16 years (sample mean), adding 2 promotional words to the abstract amounts to an additional 33.6 citations.

Can promotional language be "instrumentalized" to decrease gender bias in academia? In our analyses, there was no statistically significant evidence of promotional language being associated with a smaller gender gap in academic impact. On the contrary, when the association between promotional language and impact differed by gender, promotional language was associated with a stronger, not weaker, gender gap, with "self-promoting" men (vs. women) receiving even more citations, paper views and online mentions. This finding suggests that women are unlikely to advance by just mirroring the men's behavior[9] and that hyping might contribute to gender bias in academia. Do our findings support the gender backlash idea where

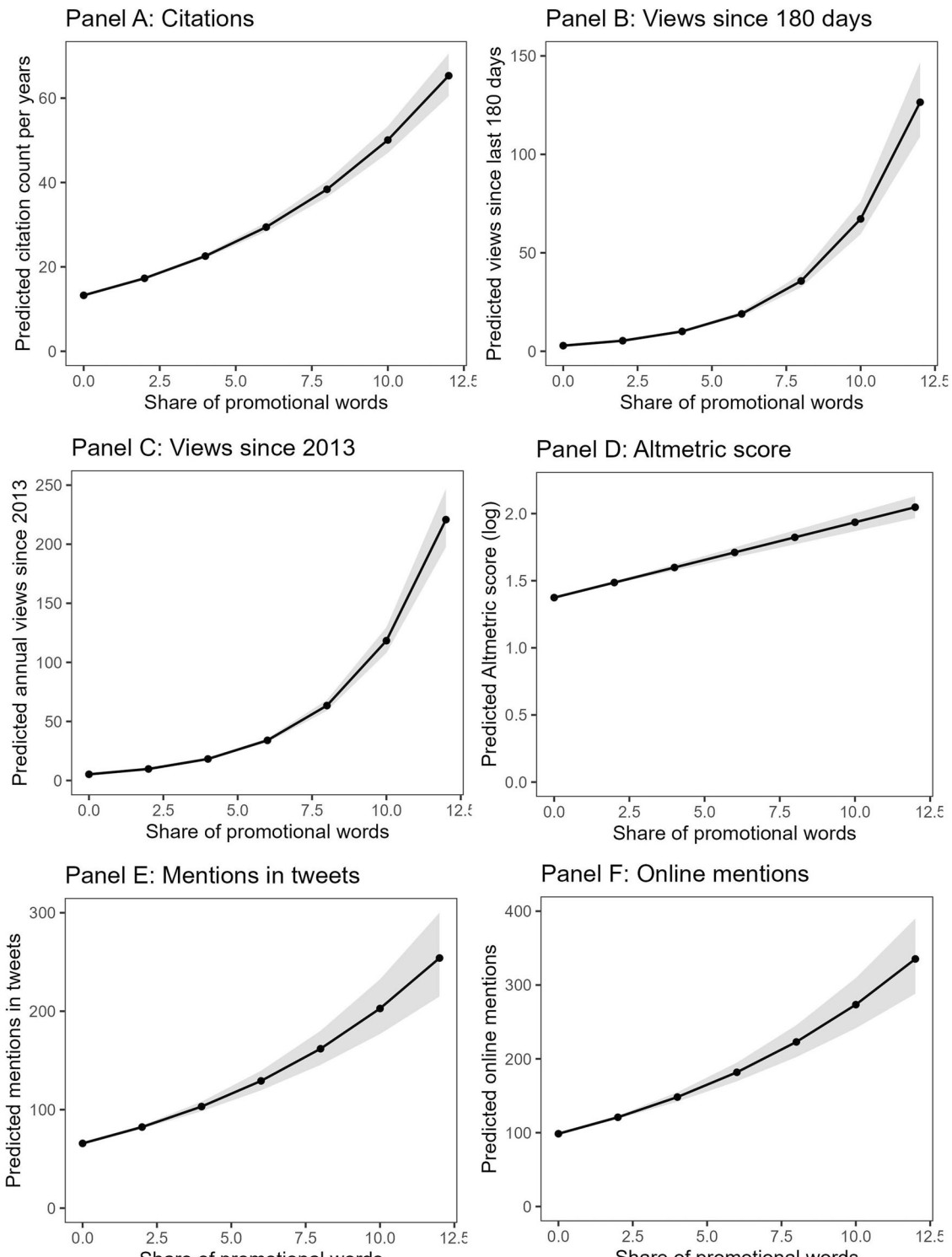

**Fig. 1 | Predicted academic impact and public attention.** *Note.* Negative binomial models were used to predict count outcomes (citations, views, mentions in tweets and posts); a linear regression model was used to predict Altmetric score. Confidence bands represent 95% CI (Panels (**A**−**C**) $n = 136{,}612$; Panels (**D**−**F**) $n = 64690$). Dots show predicted values from the models without covariates.

women adopting typical male behaviors (e.g., engaging in self-promotion rather than presenting their work in a more modest manner) experience a backlash and receive worse evaluations than men engaging in the same behaviors or women who adopt more gender-congruent behaviors[8,9]? Only partially: Only in about half of the models did we find that using promotional language is associated with more benefits (e.g., more impact) for male than for female (lead and senior) authors. Further, none of the models showed an association between self-promotion and *reduced* impact in papers led by female authors. This suggests that while the link between promotional language and higher impact may be smaller for females than for males, we found no evidence of more (vs. less) promotional language being linked to a lower impact in women. If promotional language does not correspond to reduced impact, why don't woman engage in self-promotion more (i.e., as much as men do)? While fear or backlash might be one reason,

**Table 2 | Impact and promotional language use by papers with a female (vs. male) first and last author**

| | Male | | Female | | | | | | |
|---|---|---|---|---|---|---|---|---|---|
| | *M* | *SD* | *M* | *SD* | *t* | *df* | *Cohen's d* | *95% CI* | *p* |
| **Fist author gender** | | | | | | | | | |
| Share of promotional language | 0.998 | 0.866 | 0.943 | 0.818 | 6.77 | 31632 | 0.06 | [0.05-0.08] | <0.001 |
| Citation count per year | 17.398 | 36.816 | 14.078 | 24.823 | 11.64 | 42308 | 0.10 | [0.08-0.12] | <0.001 |
| Views since last 180 days | 4.955 | 15.004 | 3.531 | 9.978 | 12.33 | 42714 | 0.10 | [0.09-0.12] | <0.001 |
| Views since 2013 per year | 9.852 | 23.913 | 7.059 | 15.554 | 15.34 | 43349 | 0.13 | [0.11-0.15] | <0.001 |
| Altmetric score | 165.005 | 480.490 | 148.472 | 451.400 | 7.92 | 20898 | 0.09 | [0.07-0.11] | <0.001 |
| Mentions in tweets | 96.271 | 504.846 | 85.962 | 569.834 | 1.63 | 19175 | 0.02 | [0-0.04] | 0.103 |
| Mentions in posts | 137.752 | 635.372 | 121.059 | 730.158 | 2.07 | 18892 | 0.03 | [0-0.05] | 0.039 |
| **Last author gender** | | | | | | | | | |
| Share of promotional language | 0.968 | 0.849 | 0.970 | 0.821 | -0.22 | 15380 | 0 | [-0.02- 0.02] | 0.828 |
| Citation count per year | 16.812 | 34.862 | 13.968 | 26.034 | 9.36 | 19179 | 0.09 | [0.06-0.11] | <0.001 |
| Views since last 180 days | 4.916 | 15.149 | 4.217 | 13.290 | 4.69 | 16573 | 0.05 | [0.03-0.07] | <0.001 |
| Views since 2013 per year | 9.785 | 24.787 | 8.111 | 19.737 | 7.39 | 18032 | 0.07 | [0.05-0.09] | <0.001 |
| Altmetric score | 152.322 | 482.327 | 146.132 | 405.309 | 0.64 | 11261 | -0.01 | [-0.03- 0.02] | 0.122 |
| Mentions in tweets | 89.175 | 549.516 | 85.697 | 457.813 | 0.56 | 12994 | -0.01 | [-0.03 -0.02] | 0.565 |
| Mentions in posts | 127.605 | 708.879 | 121.070 | 568.286 | 0.86 | 13455 | -0.01 | [-0.03 -0.02] | 0.388 |

*Note.* Altmetric score: row means, the analyses used log-transformed value.

other factors could also play a role. For example, women may fear being seen as less likable if they promote themselves too assertively[56]. Similarly, self-promotion in impression management may clash with traditional expectations about the female gender, making it appear a riskier communication strategy for women. Finally, a perspective on gendered communication suggests that unwillingness to engage in self-promotion might originate in women following others' advice. For example, studies on the feedback that women receive for scholarly submissions suggest that women are more frequently asked to use less assertive language[57]. It might also be fruitful to understand the conditions under which promotional language use by female authors is linked to more or less impact, including whether the paper is single- or multi-author or whether it was published in a discipline with a higher or lower female representation. Taken together, the potential consequences of the gendered pattern of science communication for gender diversity in academia, as well as the role of communication strategies deserve more research attention.

**Strengths, limitations and future directions**
The present analysis has a number of strengths. We included diverse indicators of impact, with some reflecting the impact achieved more recently (focusing on citations within the last six months) and others reflecting the overall impact since publication year (overall citation numbers obtained annually). Besides academic impact, we analyzed different measures of public attention, including an overall number of mentions in various online resources (social media, online news and other online documents), mentions on Twitter/X specifically and the overall Altmetric score. Further, our conclusions are based on the data regarding a large number of interdisciplinary publications (~130,000) that appeared within a long time frame (last 30 years) and were obtained from a validated source commonly used for bibliometric analysis: Web of Science[33]. It is noteworthy though that while Web of Science provides reliable data on citation count, the usage indicators (i.e., number of times a paper was accessed on the Web of Science platform) may be limited, as they only reflect access through the platform, which may not be the primary means of access for many readers. For example, including the records of view numbers through other platforms, such as Mendeley, CiteULike or GoogleScholar, might further enrich the analyses. Nevertheless, the strong positive associations between citation counts and usage indicators (see Supplementary Table 11), along with the consistency of our findings across different measures of academic impact,

suggest that although usage numbers may be underreported, this is unlikely to distort their relationship with promotional language.

The downside of our focus on available and established metrics (e.g., citations) to measure impact in academia is that these metrics are known to be affected by factors not related to the quality of the work, such as the academic discipline[58]. For example, some of the "promotional language" terms might have several meanings (e.g., "stellar" as "exceptional" vs. describing stars in astronomy) and can thus artificially inflate the promotional language score for some papers. Yet, one of our robustness checks that repeated the analyses while excluding abstracts with an exceptionally high (3 SD above the mean) share of promotional words (presented in Supplementary Tables 4-6), suggests that this is unlikely to represent a serious shortcoming. Further, our analyses controlled for a large number of potential confounding variables, including the academic discipline, and our results were robust against those controls. Still, future research could consider using field-normalized citation counts as an alternative approach to measuring impact.

Regardless of the specific form in which citation counts are used, their susceptibility to factors likely unrelated to scientific quality – such as the popularity of research topics[59] – raises the question about how well these metrics reflect scientific merit and societal value. Our findings showing that both author gender and the gendered language use matter for performance on these metrics add to this ongoing debate by suggesting that these metrics are also likely shaped by structural inequalities and gendered norms surrounding assertiveness and self-promotion. Consequently, giving these limitations of impact indicators, it might be interesting for future work to establish whether promotional language represents a kind of attention seeking behavior on the part of the authors aiming to enhance the visibility of their work, rather than reflects the work's inherent scientific qualities and the societal value.

While we used the Altmetric Attention Score as it offers a standardized indicator of public engagement with scientific publications, we also acknowledge the lack of transparency in how the score is calculated by the Altmetric Corporation. Yet, the fact that the results obtained using the Altmetric score converge with the results of two raw indicators we used – mentions in Twitter/X and the overall count of online mentions (posts) – points at the overall reliability of the findings. Nevertheless, as the online media landscape is rapidly changing (with new social media platforms emerging and others fading), future research may benefit from examining public attention separately across different platforms.

## Table 3 | Effect of promotional language moderated by first author gender

| Predictors | Citations | | | Views 180 | | | Views 2013 | | | Altmetric | | | Tweets | | | Posts | | |
|---|---|---|---|---|---|---|---|---|---|---|---|---|---|---|---|---|---|---|
| | IRR | CI | p | IRR | CI | p | IRR | CI | p | b | CI | p | IRR | CI | p | IRR | CI | p |
| (Intercept) | 15.81 | 15.53–16.10 | <0.001 | 4.15 | 4.03–4.28 | <0.001 | 8.44 | 8.26–8.62 | <0.001 | 1.49 | 1.48–1.51 | <0.001 | 87.50 | 84.71–90.38 | <0.001 | 125.10 | 121.39–128.92 | <0.001 |
| Promotional language | 1.10 | 1.08–1.11 | <0.001 | 1.18 | 1.16–1.21 | <0.001 | 1.16 | 1.14–1.18 | <0.001 | 0.05 | 0.04–0.06 | <0.001 | 1.09 | 1.07–1.12 | <0.001 | 1.09 | 1.07–1.12 | <0.001 |
| First author gender | 0.83 | 0.81–0.86 | <0.001 | 0.72 | 0.69–0.76 | <0.001 | 0.73 | 0.70–0.75 | <0.001 | -0.08 | -0.11–-0.05 | <0.001 | 0.93 | 0.87–0.98 | 0.009 | 0.92 | 0.87–0.97 | 0.002 |
| Promotional language * First author gender | 0.98 | 0.95–1.00 | 0.070 | 0.99 | 0.95–1.04 | 0.809 | 1.00 | 0.97–1.03 | 0.779 | 0.00 | -0.02–-0.03 | 0.735 | 0.97 | 0.93–1.01 | 0.190 | 0.96 | 0.92–1.00 | 0.069 |
| Observations | 48241 | | | 48241 | | | 48241 | | | 35209 | | | 35209 | | | 35209 | | |
| R² | 0.019 | | | 0.022 | | | 0.033 | | | 0.005 / 0.004 | | | 0.004 | | | 0.005 | | |
| (Intercept) | 16.58 | 15.48–17.75 | <0.001 | 0.19 | 0.17–0.21 | <0.001 | 0.33 | 0.30–0.36 | <0.001 | 0.11 | 0.05–0.18 | 0.001 | 0.38 | 0.33–0.43 | <0.001 | 2.31 | 2.05–2.61 | <0.001 |
| Promotional language | 1.08 | 1.07–1.10 | <0.001 | 1.12 | 1.10–1.14 | <0.001 | 1.11 | 1.09–1.12 | <0.001 | 0.04 | 0.03–0.05 | <0.001 | 1.05 | 1.03–1.07 | <0.001 | 1.06 | 1.04–1.08 | <0.001 |
| First author gender | 0.93 | 0.90–0.95 | <0.001 | 0.82 | 0.78–0.86 | <0.001 | 0.84 | 0.81–0.86 | <0.001 | -0.04 | -0.06–-0.01 | 0.003 | 0.92 | 0.88–0.97 | 0.002 | 0.90 | 0.86–0.94 | <0.001 |
| Journal [nature] | 3.27 | 3.20–3.34 | <0.001 | 3.56 | 3.43–3.70 | <0.001 | 3.87 | 3.78–3.97 | <0.001 | 0.80 | 0.78–0.82 | <0.001 | 3.44 | 3.31–3.57 | <0.001 | 3.53 | 3.40–3.66 | <0.001 |
| Journal [science] | 3.27 | 3.18–3.36 | <0.001 | 4.38 | 4.18–4.58 | <0.001 | 4.74 | 4.60–4.89 | <0.001 | 0.86 | 0.83–0.88 | <0.001 | 3.72 | 3.54–3.90 | <0.001 | 3.84 | 3.67–4.02 | <0.001 |
| Year | 0.97 | 0.96–0.97 | <0.001 | 1.09 | 1.09–1.10 | <0.001 | 1.08 | 1.08–1.09 | <0.001 | 0.05 | 0.05–0.06 | <0.001 | 1.22 | 1.22–1.22 | <0.001 | 1.15 | 1.15–1.16 | <0.001 |
| Word count | 1.00 | 1.00–1.00 | <0.001 | 1.00 | 1.00–1.00 | <0.001 | 1.00 | 1.00–1.00 | <0.001 | 0.00 | 0.00–0.00 | <0.001 | 1.00 | 1.00–1.00 | <0.001 | 1.00 | 1.00–1.00 | <0.001 |
| Field [formal & natural] | 0.82 | 0.79–0.86 | <0.001 | 0.85 | 0.79–0.90 | <0.001 | 1.02 | 0.98–1.07 | 0.304 | -0.59 | -0.62–-0.56 | <0.001 | 0.23 | 0.22–0.24 | <0.001 | 0.27 | 0.26–0.29 | <0.001 |
| Field [medical & health] | 0.99 | 0.95–1.04 | 0.802 | 0.54 | 0.51–0.58 | <0.001 | 0.68 | 0.64–0.71 | <0.001 | -0.47 | -0.51–-0.44 | <0.001 | 0.43 | 0.40–0.46 | <0.001 | 0.47 | 0.44–0.50 | <0.001 |
| Author number | 1.01 | 1.01–1.01 | <0.001 | 1.01 | 1.01–1.01 | <0.001 | 1.01 | 1.01–1.01 | <0.001 | 0.00 | 0.00–0.00 | <0.001 | 1.01 | 1.01–1.01 | <0.001 | 1.01 | 1.01–1.01 | <0.001 |
| Institutional diversity | 0.91 | 0.90–0.93 | <0.001 | 1.08 | 1.04–1.11 | <0.001 | 1.06 | 1.04–1.08 | <0.001 | 0.08 | 0.07–0.10 | <0.001 | 1.38 | 1.34–1.43 | <0.001 | 1.28 | 1.24–1.32 | <0.001 |
| Positivity | 1.00 | 1.00–1.00 | <0.001 | 1.01 | 1.00–1.00 | <0.001 | 1.01 | 1.01–1.01 | <0.001 | 0.00 | 0.00–0.00 | <0.001 | 1.00 | 1.00–1.00 | <0.001 | 1.00 | 1.00–1.00 | <0.001 |
| Promotional language * First author gender | 0.98 | 0.96–1.00 | 0.032 | 1.01 | 0.97–1.04 | 0.716 | 1.00 | 0.98–1.03 | 0.834 | 0.01 | -0.01–-0.03 | 0.453 | 0.94 | 0.91–0.98 | 0.002 | 0.95 | 0.92–0.99 | 0.008 |
| Observations | 48241 | | | 48241 | | | 48241 | | | 35209 | | | 35209 | | | 35209 | | |
| R² | 0.536 | | | 0.386 | | | 0.613 | | | 0.305 / 0.304 | | | 0.581 | | | 0.530 | | |

*Note. b* = unstandardized regression coefficient; IRR = Incidence Rate Ratio; CI = 95% Confidence Interval. Citations: citation count per year; Views 180: the number of times the record has been accessed on the Web of Science platform (e.g. by clicking on the link to the publisher's website or by saving it in a bibliographic management tool) in the last 180 days; Views 2013: the number of times the record has been accessed on the Web of Science platform (e.g. by clicking on the link to the publisher's website or by saving it in a bibliographic management tool) since 2013, per year. Altmetric: Altmetric score; Tweets: number of mentions on Twitter/X; Posts: overall number of mentions in any type of online document (news, social media, policy documents etc.). First author gender: 1 = female, 0 = male. Promotional language: share of promotional language in the abstract; Journal: PNAS is reference category; Year: publication year; Word count: number of words in the abstract; Field: social and humanities is reference category; author number: number of authors; institutional diversity: number of listed affiliations divided by the number of authors, higher score indicates more collaborations across different institutions; Positivity: overall semantic positivity of the abstract, obtained using the Linguistic Inquiry and Word Count software (LIWC, version 2022), R²: for negative binomial models, Nagelkerke's R² is reported, for a linear model (Altmetric sore), multiple R²/adjusted R² are reported.

**Table 4 | Effect of promotional language moderated by last author gender**

| Predictors | Citations | | | Views 180 | | | Views 2013 | | | Altmetric | | | Tweets | | | Posts | | |
|---|---|---|---|---|---|---|---|---|---|---|---|---|---|---|---|---|---|---|
| | IRR | CI | p | IRR | CI | p | IRR | CI | p | b | CI | p | IRR | CI | p | IRR | CI | p |
| (Intercept) | 15.28 | 15.05 – 15.51 | <0.001 | 4.04 | 3.94 – 4.14 | <0.001 | 8.28 | 8.13 – 8.43 | <0.001 | 1.44 | 1.42 – 1.45 | <0.001 | 81.30 | 79.12 – 83.54 | <0.001 | 116.02 | 113.12 – 118.99 | <0.001 |
| Promotional language | 1.10 | 1.09 – 1.11 | <0.001 | 1.21 | 1.18 – 1.23 | <0.001 | 1.18 | 1.16 – 1.19 | <0.001 | 0.06 | 0.05 – 0.07 | <0.001 | 1.09 | 1.07 – 1.12 | <0.001 | 1.10 | 1.08 – 1.12 | <0.001 |
| Last author gender | 0.88 | 0.85 – 0.91 | <0.001 | 0.93 | 0.88 – 0.99 | 0.020 | 0.87 | 0.83 – 0.91 | <0.001 | 0.03 | -0.00 – 0.06 | 0.059 | 1.01 | 0.95 – 1.08 | 0.783 | 1.00 | 0.94 – 1.06 | 0.923 |
| Promotional language * Last author gender | 0.95 | 0.92 – 0.98 | <0.001 | 0.93 | 0.88 – 0.97 | 0.002 | 0.96 | 0.93 – 0.99 | 0.015 | -0.04 | -0.06 – -0.01 | 0.003 | 0.95 | 0.91 – 1.00 | 0.066 | 0.95 | 0.91 – 1.00 | 0.045 |
| Observations | 56824 | | | 56824 | | | 56824 | | | 41455 | | | 41455 | | | 41455 | | |
| Nagelkerke's R² | 0.013 | | | 0.013 | | | 0.019 | | | 0.003 | | | 0.003 | | | 0.004 | | |
| (Intercept) | 16.27 | 15.26 – 17.34 | <0.001 | 0.16 | 0.14 – 0.18 | <0.001 | 0.28 | 0.26 – 0.31 | <0.001 | 0.15 | 0.08 – 0.21 | <0.001 | 0.24 | 0.21 – 0.27 | <0.001 | 1.74 | 1.56 – 1.95 | <0.001 |
| Promotional language | 1.09 | 1.08 – 1.10 | <0.001 | 1.14 | 1.12 – 1.16 | <0.001 | 1.12 | 1.11 – 1.13 | <0.001 | 0.05 | 0.04 – 0.06 | <0.001 | 1.05 | 1.03 – 1.07 | <0.001 | 1.07 | 1.05 – 1.08 | <0.001 |
| Last author gender | 0.97 | 0.94 – 1.00 | 0.064 | 0.96 | 0.91 – 1.01 | 0.137 | 0.94 | 0.90 – 0.97 | <0.001 | 0.04 | 0.02 – 0.07 | 0.001 | 0.86 | 0.82 – 0.91 | <0.001 | 0.91 | 0.87 – 0.96 | <0.001 |
| Journal [nature] | 3.30 | 3.23 – 3.36 | <0.001 | 3.65 | 3.53 – 3.78 | <0.001 | 4.05 | 3.96 – 4.15 | <0.001 | 0.81 | 0.79 – 0.83 | <0.001 | 3.54 | 3.42 – 3.67 | <0.001 | 3.55 | 3.43 – 3.67 | <0.001 |
| Journal [science] | 3.43 | 3.35 – 3.52 | <0.001 | 4.73 | 4.53 – 4.93 | <0.001 | 5.19 | 5.04 – 5.34 | <0.001 | 0.86 | 0.84 – 0.89 | <0.001 | 4.13 | 3.95 – 4.32 | <0.001 | 4.08 | 3.91 – 4.25 | <0.001 |
| Year | 0.96 | 0.96 – 0.97 | <0.001 | 1.09 | 1.09 – 1.10 | <0.001 | 1.09 | 1.08 – 1.09 | <0.001 | 0.05 | 0.05 – 0.05 | <0.001 | 1.23 | 1.23 – 1.23 | <0.001 | 1.16 | 1.15 – 1.16 | <0.001 |
| Word count | 1.00 | 1.00 – 1.00 | <0.001 | 1.00 | 1.00 – 1.00 | <0.001 | 1.00 | 1.00 – 1.00 | <0.001 | 0.00 | 0.00 – 0.00 | <0.001 | 1.00 | 1.00 – 1.00 | <0.001 | 1.00 | 1.00 – 1.00 | <0.001 |
| Field [formal & natural] | 0.82 | 0.79 – 0.85 | <0.001 | 0.87 | 0.82 – 0.92 | <0.001 | 1.05 | 1.01 – 1.10 | 0.026 | -0.60 | -0.63 – -0.57 | <0.001 | 0.22 | 0.21 – 0.24 | <0.001 | 0.27 | 0.25 – 0.28 | <0.001 |
| Field [medical & health] | 0.97 | 0.93 – 1.01 | 0.108 | 0.53 | 0.49 – 0.56 | <0.001 | 0.66 | 0.63 – 0.69 | <0.001 | -0.50 | -0.53 – -0.46 | <0.001 | 0.39 | 0.37 – 0.42 | <0.001 | 0.44 | 0.41 – 0.47 | <0.001 |
| Author number | 1.02 | 1.02 – 1.02 | <0.001 | 1.01 | 1.01 – 1.01 | <0.001 | 1.01 | 1.01 – 1.01 | <0.001 | 0.00 | 0.00 – 0.00 | <0.001 | 1.01 | 1.01 – 1.01 | <0.001 | 1.01 | 1.01 – 1.01 | <0.001 |
| Institutional diversity | 0.94 | 0.92 – 0.95 | <0.001 | 1.08 | 1.05 – 1.11 | <0.001 | 1.06 | 1.04 – 1.08 | <0.001 | 0.09 | 0.07 – 0.10 | <0.001 | 1.50 | 1.46 – 1.55 | <0.001 | 1.37 | 1.34 – 1.41 | <0.001 |
| Positivity | 1.00 | 1.00 – 1.00 | <0.001 | 1.01 | 1.01 – 1.01 | <0.001 | 1.01 | 1.01 – 1.01 | <0.001 | 0.00 | 0.00 – 0.00 | <0.001 | 1.00 | 1.00 – 1.00 | <0.001 | 1.00 | 1.00 – 1.00 | <0.001 |
| Promotional language * Last author gender | 0.97 | 0.94 – 0.99 | 0.004 | 0.98 | 0.94 – 1.02 | 0.400 | 1.00 | 0.97 – 1.03 | 0.938 | -0.03 | -0.05 – -0.00 | 0.015 | 0.99 | 0.95 – 1.03 | 0.610 | 0.97 | 0.93 – 1.01 | 0.132 |
| Observations | 56824 | | | 56824 | | | 56824 | | | 41455 | | | 41455 | | | 41455 | | |
| R² | 0.545 | | | 0.408 | | | 0.637 | | | 0.308 / 0.308 | | | 0.593 | | | 0.530 | | |

*Note. b = unstandardized regression coefficient; IRR = Incidence Rate Ratio; CI = 95% Confidence Interval. Citations: citation count per year; Views 180: the number of times the record has been accessed on the Web of Science platform (e.g. by clicking on the link to the publisher's website or by saving it in a bibliographic management tool) in the last 180 days; Views 2013: the number of times the record has been accessed on the Web of Science platform (e.g. by clicking on the link to the publisher's website or by saving it in a bibliographic management tool) since 2013, per year. Altmetric: Altmetric score; Tweets: number of mentions on Twitter/X; Posts: overall number of mentions in any type of online document (news, social media, policy documents etc.). Last author gender: 1 = female, 0 = male. Promotional language: share of promotional language in the abstract; Journal: PNAS is reference category; Year: publication year; Word count: number of words in the abstract; Field: social and humanities is reference category; author number: number of authors; institutional diversity: number of listed affiliations divided by the number of authors, higher score indicates more collaborations across different institutions; Positivity: overall semantic positivity of the abstract, obtained using the Linguistic Inquiry and Word Count software (LIWC, version 2022). R²: for negative binomial models, Nagelkerke's R² is reported, for a linear model (Altmetric sore), multiple R²/adjusted R² are reported.*

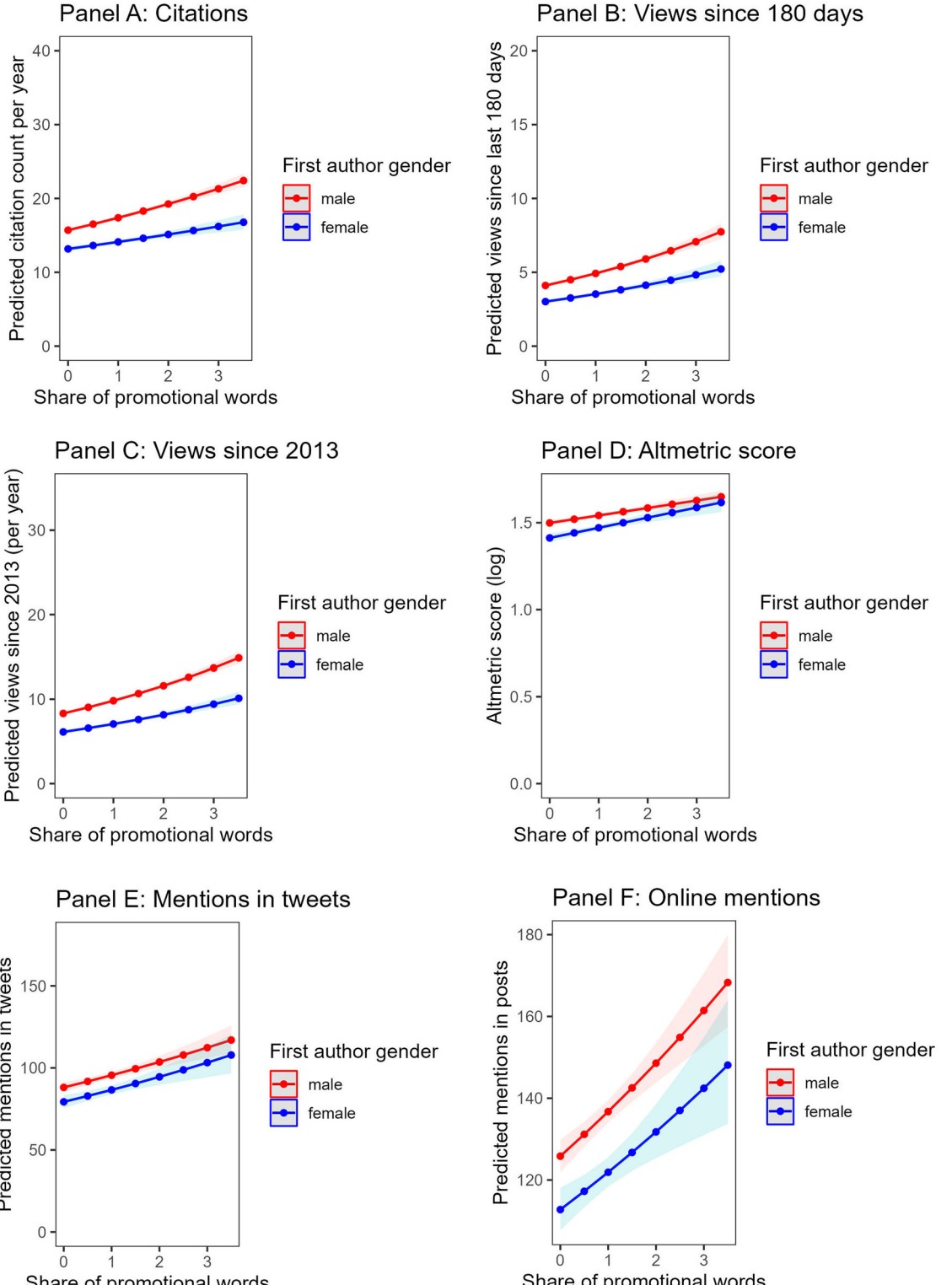

**Fig. 2 | Predicted academic impact and public attention of papers with a male vs. a female first author.** *Note* Negative binomial models were used to predict count outcomes (citations, views, mentions in tweets and posts); a linear regression model was used to predict Altmetric score. Confidence bands represent 95% CI (Panels (**A**−**C**) $n = 48,241$; Panels (**D**−**F**) $n = 35,209$). Dots show predicted values from the models without covariates.

Following prior work[1,23,24], we relied on a dictionary approach to measure promotional language. One limitation of the dictionary approach to text analysis is that it relies on predefined word lists, which may overlook context, nuance, and evolving language use. Additionally, such approaches can mis-categorize words with multiple meanings and fail to capture the broader semantic structure of a text. Although our robustness checks indicate that this issue is unlikely to have driven our findings, we encourage future research to adopt more advanced text analysis methods that are better equipped to handle such limitations – such as distributed dictionary representation (DDR), which enables a more semantically nuanced measurement based on an initial dictionary[60].

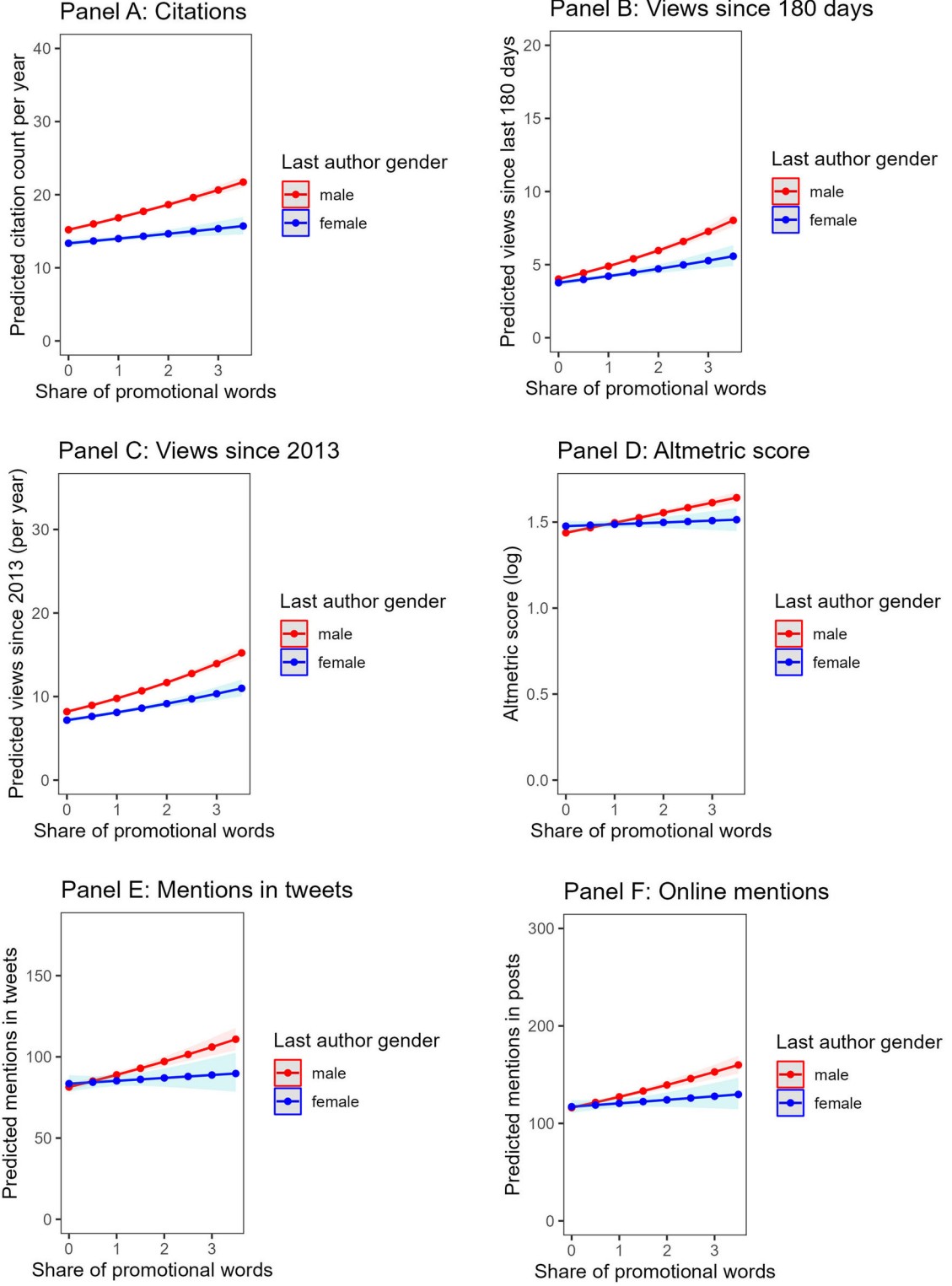

**Fig. 3 | Predicted academic impact and public attention of papers with a male vs. a female last author.** *Note* Negative binomial models were used to predict count outcomes (citations, views, mentions in tweets and posts); a linear regression model was used to predict Altmetric score. Confidence bands represent 95% CI (Panels (**A–C**) *n* = 56,824; Panels (**D–F**) *n* = 41,455). Dots show predicted values from the models without covariates.

Another important limitation pertains to the gender detection from first names. Due to the impracticality of managing large sample sizes with human raters, algorithmic gender detection is a widely used approach in bibliometric analyses[28,33]. Indeed, an additional validation analysis of a selection of first names revealed an extremely high agreement between the automatic detection algorithm and a human rater. Yet, gender classification based on first names (when done by humans or algorithms) is often problematic in the context of Asian names, as they are frequently androgynous (i.e., commonly used by both males and females)[61]. As a result of this limitation, the algorithm was unable to provide classifications for a significant portion of the dataset. Consequently, our gender analyses were limited, meaning the observed patterns likely reflect trends associated with

Western names. We encourage future studies to devote more attention to automated gender disambiguation efforts, potentially using other gender detection methods (e.g., evaluating authors' university homepages[62]) to ensure the methodological inclusivity and the classification accuracy across cultures. These efforts would allow future research to examine whether the gendered pattern of the association between promotional language and academic impact is restricted to Western countries. Also, we focused on the gender of the first and the last author here, as these roles are typically viewed as the most influential and indicative of major contributions. However, future research could apply the CRediT taxonomy to identify the gender of authors primarily responsible for writing the manuscript and examine whether similar patterns emerge in those analyses. Finally, while the present analyses centered on gender, other identities, such as race, region or institutional prestige, may shape both the use of promotional language and its association with academic impact, and should be explored in future studies.

Although a key strength of our work lies in the use of real-world data capturing the behavior of millions of scientists and consumers of science, the correlational nature of our analyses precludes drawing causal conclusions. Although we made extensive efforts to account for a wide range of potential confounding variables—such as publication year, subject field, author number and diversity, and the semantic positivity of the abstract—it is still possible that our results are driven by the possibility that papers with more promotional language use more rigorous methods and offer more reliable conclusions, are more likely to present (statistically and conceptually) significant findings as well as novel and innovative content, thereby attracting greater academic and public attention. Indeed, Peng et al. have shown that grant proposals that use promotional language tend to include citations from more diverse disciplines, speaking to these proposals' higher innovative potential[24]. At the same time, the effect of promotional language on impact might be causal for a number of reasons. First, prior work has shown that experimental manipulations of doubtful language in science communication messages leads to greater public engagement[22,63]. Second, drawing on previous research on the role of doubt in (science) communication and user engagement[19,63], there are theoretical grounds to suggest that the use of promotional language enhances impact by amplifying perceptions of novelty and importance—key factors influencing academic success. We nevertheless strongly encourage future work to consider experimental study designs that would allow to draw causal conclusion or using alternative indicators of work quality (e.g., evaluations during peer review), testing whether authors can boost their work's impact by simply using more promotional language.

Further, while we analyzed the content of three high-impact interdisciplinary journals, covering a broad range of disciplines, from mathematics, to biology, to archeology, to health sciences, future studies might explore whether promotional language plays a similar role in disciplines that are less well-represented in the journals analyzed here (e.g., humanities), as well as in less international and potentially lower-impact journals. Indeed, in management studies, the rapid increase in the use of positive language in recent years was restricted to high impact journals, where it was also linked to a higher number of citations (relative to lower impact journals)[29].

Our findings further show that the relationship between promotional language and impact has evolved over the past three decades. While its link to citation counts has declined, its association with other impact indicators—such as views and public attention—has increased over time. We speculate that the interpretation of promotional language has evolved differently across the academic audience vs. the public. For researchers deciding whether to cite a paper, the use of such language may now be met with greater skepticism – particularly in the aftermath of the replicability crisis and the credibility revolution – leading to a weakened association with citation counts. In contrast, this skepticism is less likely to influence metrics such as view count or public attention. In fact, in today's highly competitive academic environment, where research must compete for visibility, the use of promotional language may remain effective—and potentially even rise in importance—for capturing public interest and driving engagement, such as sharing on social media. Exploring how promotional language in research

communication is perceived and interpreted by different audiences could help future research understand these temporal developments.

While the use of promotional language in academic writing (and its downstream consequences) might be affected by time-related norms and fads, other very recent developments – in particular, the increasing role of AI in supporting scientific work, such as manuscript preparation and editing – might shape academic writing as well. It remains to be seen whether AI-supported editing increases the use of promotional language, potentially accelerating the visibility of such papers. This raises important questions about the broader impact of AI on scientific communication and its consequences for the academic community and society at large.

## Conclusions

In summary, is promotional language beneficial? For authors and publishers, it is: using promotional language is associated more academic impact and public attention. The benefits of promotional language for science itself and the public are however less clear. A pervasive view among meta-researchers is that it could lead to a "loss of objectivity"[64], exaggeration and sensationalization of the results, ultimately undermining reproducibility and threatening public trust in science[2,4,5,65,66]. Similarly, open science advocates and policy analysis researchers have repeatedly highlighted the importance of acknowledging the uncertainties and limitations of published research findings[67–69] – practices rather incompatible with hyping. Studies measuring the quality of published work using other tools than citation metrics, are needed to understand the potentially complicated nature of the relationship between hyping and research quality and novelty.

## Data availability

Data can be accessed at: https://osf.io/z3x4c/. https://doi.org/10.17605/OSF.IO/Z3X4C.

## Code availability

Analyses scripts can be accessed at: https://osf.io/z3x4c/. https://doi.org/10.17605/OSF.IO/Z3X4C.

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

## Acknowledgements

Anthony M. Evans is employed at Allstate Inc. and Milena Ivanovic is employed at Ipsos marketing. Their respective employers had no role in the present study. No funding was received for this work.

## Author contributions

Olga Stavrova: conceptualization, methodology, formal analysis, data curation, writing – original draft, project administration. Bennett Kleinberg:conceptualization, methodology, writing – review and editing. Anthony M. Evans: conceptualization, methodology, writing – review and editing. Milena Ivanovic: data curation.

## Funding

## Competing interests

The authors declare no competing interests.
