## [Transparent Peer Review file · Communications Psychology]

Scientific publications that use promotional language in the abstract receive more citations and public attention

Corresponding Author: Professor Olga Stavrova

Version 0:

Decision Letter:

Dear Professor Stavrova,

Thank you for your patience during the peer-review process. We apologise for the long delay in returning to you with a decision, which resulted from delays in reviewer availability. Your manuscript titled "Scientific publications that use promotional language receive more citations and social media mentions" has now been seen by 3 reviewers, and I include their comments at the end of this message. They find your work of interest but raised some important points. We are interested in the possibility of publishing your study in Communications Psychology, but would like to consider your responses to these concerns and assess a revised manuscript before we make a final decision on publication.

We therefore invite you to revise and resubmit your manuscript, along with a point-by-point response to the reviewers. Please highlight all changes in the manuscript text file.

Editorially, we consider it important to address the limitations raised related to the automated name-based gender classification approach, as the classification of author gender is a key part of the manuscript. We also emphasize that our guidelines do not support presentation of correlational evidence as causal and ask you to remove causal or mechanistic claims.

I am attaching an Editorial Requests Table that details critical reporting requirements for the revised manuscript. Please attend to each item and ensure your manuscript is fully compliant. If your revised manuscript is not aligned with these requests on major issues, such as those concerning statistics, it may be returned to you for further revisions without re-review.

Please submit the following items:

- Revised manuscript
- Point-by-point response to the referees' comments
- Cover letter (as a separate document)
- <https://www.nature.com/documents/nr-reporting-summary.zip>>Nature Research Reporting Summary
- <https://www.nature.com/documents/nr-editorial-policy-checklist.pdf>>Editorial Policy Checklist
- Completed Editorial Request Table (attached).

via this link: Link Redacted .

Additional guidance is available in our style and formatting guide Communications Psychology formatting guide.

Best regards,

Daniel Quintana

Daniel Quintana, PhD
Editorial Board Member
Communications Psychology
orcid.org/0000-0003-2876-0004

REVIEWER EXPERTISE:

Reviewer #1: Bibliometrics

Reviewer #2: Bibliometrics, Altmetrics

Reviewer #3: Altmetrics

REVIEWER REPORTS:

Reviewer #1 (Remarks to the Author):

The paper examines how promotional language in scientific publications influences academic impact and public engagement, using a dataset of over 130,000 abstracts from Science, Nature, and PNAS (1991–2023). It finds that promotional language is associated with higher citation rates, increased full-length views, and greater Altmetric scores, as well as enhanced public engagement through social media and blog mentions. The study also highlights gender disparities, showing that female scientists use promotional language less frequently and do not benefit from it as much as their male counterparts.

Strengths:

- + Large dataset covering three decades and high-impact journals.
- + Multiple impact metrics (citations, full-text views, Altmetric scores).
- + Robust statistical methods with different statistical measures.

Weaknesses:

- Correlation does not imply causation. While the study shows a link between promotional language and impact, it does not establish causality. Factors like research quality may influence both promotional language use and citation counts. Experimental testing could strengthen conclusions.
- Papers with stronger findings may naturally use more positive language, while those with less significant or negative results may tone down claims. It remains unclear whether promotional language itself drives impact or if it simply reflects the nature of the results.
- The study uses automated name-based gender classification, which can misclassify non-binary individuals and certain cultural names. A more refined classification method would improve accuracy.
- The study focuses on only three high-impact journals (Science, Nature, PNAS). This excludes disciplinary differences and journals with different editorial styles. Expanding to a wider range of journals would improve generalizability.

While this paper contributes to ongoing discussions on science communication and linguistic framing, similar research exists.

1. Edlinger et al. (2023)

Found that positive language in abstracts has increased and is strategically placed to influence perception. Given this, how does the current study add to the understanding beyond showing citation benefits?

Edlinger, M., Buchrieser, F., & Wood, G. (2023). Presence and consequences of positive words in scientific abstracts. *Scientometrics*, 128(12), 6633-6657.

2. Lerchenmueller et al. (BMJ 2019)

Found that positive framing in scientific abstracts correlates with higher citation counts and that male scientists use promotional language more often. How does this study advance beyond confirming these findings?

Lerchenmueller, M. J., Sorenson, O., & Jena, A. B. (2019). Gender differences in how scientists present the importance of their research: observational study. *BMJ*, 367.

3. Wu et al. (2024)

Found that positive words predict higher citation counts but noted a declining citation advantage over time. In contrast, this study suggests that promotional language continues to boost citations and engagement. Could the authors explain this discrepancy?

Wu, D., Wu, H., & Li, J. (2024). Citation advantage of positive words: predictability, temporal evolution, and universality in varied quality journals. *Scientometrics*, 129(7), 4275-4293.

Reviewer #2 (Remarks to the Author):

This study examines the potential effects of promotional language in scientific publications published in three iconic venues (Science, Nature and PNAS). The authors rely on specific terms in the publications' abstracts to do this.

Overall, the work presents objectives of great interest to the meta-research community and quantitative studies in science. The manuscript is very well structured and written. The methodology followed especially the statistical component, is robust and provides highly interesting results for understanding the effect of the use of populist terms on the impact of the papers. The conclusions are also derived from the data and are nuanced and contextualized. For this reason, I am positive about the publication of this work.

However, I note some issues that I will address below in order to improve the justification of some decisions and enhance the manuscript.

Introduction

On line 131, the authors indicate "across over 50,000 publications." However, I cannot see where this number of publications is, as 130,000 abstracts were analyzed after the filtering process.

As regards the gender issue, the authors introduce the topic indicating that "There is one factor that is particularly likely to affect the effectiveness of promotional language: author gender". However, from my point of view, the gender issue should be introduced in the study in a slightly different way, specifically as one, among many others, of the potential consequences of promotional language. The question is whether promotional language is more effective according to the author's gender, considering that women tend, on average, to use less promotional language. Consequently, the promotional language should be an additional barrier to gaining citations and visibility.

Method

2023 is a very recent publication to consider the number of citations, especially considering that data was collected by June 2023. The authors indicate this issue later on lines 209 and 210, mentioning the altmetrics collected. Publications before 2011/2012 cannot be compared to other recent publications.

Otherwise, the authors use the Altmetric attention score. However, this indicator has been described in the literature as an opaque and non-responsible metric not recommended for bibliometric studies. Authors should justify using this composed indicator better than raw metrics. In this sense, the number of tweets is one of the most important altmetric indicators. Still, the number of Mendeley readers could have provided other interesting findings and the presence of publications in news and press releases.

While the authors use several metrics to control de variability, I suggest that future studies collect field-normalized citation-based data.

Results

I probably misunderstood the logic, but the authors say, “last author gender was not significantly associated with the share of promotional words” (lines 242/243). However, the authors say later, “Again, the impact gap between female- and male-last author papers increased with increasing promotional language” (lines 256/257). Later, in Figure 2 panel B, we can observe significant differences for the last author in all control metrics, including sharing promotional words. It was somewhat confusing to follow the authors’ argument.

Discussion

As the authors already indicate, causal conclusions cannot be driven. In addition, the quality or novelty of papers should not be discussed here. No assumption should be made about using promotional language to indicate that the study is less serious, especially when analyzing only the abstract of publications. Abstracts can be written to catch the readers (sometimes even requested by the journals), but the full-text manuscript can be accurate.

Moreover, the study covered three special multidisciplinary journals, with huge coverage and strong communication activities, and was frequently mentioned in the press. I don’t think that these journals can represent the scientific activity. These aren’t “normal journals”, bibliometrically speaking. A comment on this issue is necessary, in my opinion.

Otherwise, the authors do not mention other studies focused on using promotional language in the titles that can add background to this manuscript. In this same line, I miss a discussion on the effect of AI tools when writing publications. Literature has already evidenced that specific terms are overused due to AI tools that can affect visibility and readability.

Finally, future studies should use credit taxonomy to analyze the effects of promotional language by author gender, considering the person with the credit for writing the manuscript, regardless of their position.

Minor issues:

Please, fix some typos: “assertive language” “reent study” “scientsts”

Keywords should be improved, as relevant terms are omitted, such as gender analysis or scientific promotion.

References 21 and 22 seem the same. Please check.

Reviewer #3 (Remarks to the Author):

This manuscript offers a large-scale and methodologically sound analysis of over 130,000 scientific abstracts, examining whether the use of promotional language is associated with greater academic and public impact. The findings are compelling: promotional language is positively associated with citation counts and media visibility, but the effects are not equally distributed across gender lines. Specifically, promotional language tends to amplify the existing gender citation and visibility gap rather than mitigate it, with male authors benefiting more than their female counterparts. This work is important and timely. The authors tackle a subject at the intersection of metascience, communication psychology, and gender studies, offering empirical clarity on a cultural practice that has long simmered under anecdotal and editorial scrutiny. We commend the authors for their ambitious scope and analytical rigor. That said, we would like to see more interpretive clarity and a deeper discussion, particularly around gender inference, the normativity of “impact,” and the operationalization of promotional language. These areas would benefit from further reflection.

Major comments:

1. The meaning and normativity of “impact”

The authors present impact as an inherently desirable outcome (e.g., more citations, higher Altmetric scores). We understand that citations are the de facto currency in academia, but would encourage the authors to reflect more critically on what these metrics actually signify. To what extent do citations or tweet counts indicate scientific quality or social value, and to what extent do they simply reflect success within the attention economy, shaped by performative cues and systemic inequities? This question becomes especially salient in light of the gendered dynamics at play: if men are rewarded more for using promotional language, are we witnessing meritocratic advantage, or a socially-coded responsiveness to assertive (often male-coded) behaviors? It might strengthen the discussion to include a brief reflection on the value-neutrality (or lack thereof) of “impact” metrics, and what that means for interpreting the findings.

2. Operationalization of promotional language

The dictionary-based approach is reasonable and well-situated within prior literature. However, we are wondering whether the paper engages deeply enough with the issue of polysemy and disciplinary norms. Words like novel or robust may be promotional in one context but simply technical in another. Could some disciplines (e.g., astrophysics vs. medicine) be systematically flagged as “more promotional” due to these differences? We’d be curious to hear whether the authors tested for dictionary validity across time or field, and whether certain domains were disproportionately flagged for high promotional word use.

3. Gender inference and cultural bias

The authors rightly note the limitations of using name-based gender classification, particularly for Asian names. However, the claim that this ambiguity may “protect” some authors from gender bias feels speculative and potentially problematic.

Framing it that way risks naturalizing a Western-centric lens on name recognition and gender signaling. Rather than describing gender ambiguity as a protective shield, perhaps frame it as a gap in methodological inclusivity, one that future work could address through cross-referencing institutional data or author self-identification. Additionally, we'd encourage the authors to acknowledge that race, region, institutional prestige, and other intersecting identities may also shape both promotional strategies and their downstream effects. These dynamics deserve more attention, even if they are beyond the scope of the current dataset.

4. The backlash hypothesis

The authors gesture toward the backlash hypothesis but also acknowledge that the findings only partially support it. We appreciated this nuance, though at times the conclusion seems to stretch slightly beyond what the data warrant. Observing smaller gains for women is not the same as showing that self-promotion hurts women. But it certainly invites questions about why women may (or may not) engage in self-promotion at the same rate as men. This could be a good opportunity to incorporate perspectives from impression formation or gendered communication research, which might offer alternative explanations beyond backlash, such as strategic ambiguity, likability penalties, or status expectation mismatches.

Minor comments:

5. Does all promotional language count as "hying?" The paper could do more than providing a few example keywords to train the reader's intuition about what kind of self-promotional language is studied.
6. The manuscript comments on possible issues with the relevancy of views (i.e., the number of times the paper's record has been accessed on the Web of Science platform). It seems that the two analyses based on views are robustness checks, complementing the investigation focused on citations.
7. When the authors are quantifying citations, they use the yearly average citations even though we know that citations are not accumulated linearly. The paper could acknowledge this possible shortcoming in measurement.
8. The Altmetric score includes mentions in blogposts and on Twitter. As the paper acknowledges, these three indicators are somewhat correlated, so here too it may be worth showing one of them and use the others as robustness checks. The current title could guide the selection of what may constitute the main analysis.
9. Please mention the rationale behind recoding of the 20 fields and list which fields belong to the three aggregated groups.
10. Can you please specify the fit of the models in Table 1?
11. It seems prudent to stay away from suggestions like "testing whether authors can boost the impact of lower-quality work simply by using more promotional language."

This is a compelling and necessary paper. It offers strong empirical evidence for an effect that many have intuited but few have rigorously demonstrated: that promotional language boosts visibility—but not equitably. We found the gender moderation analysis particularly thought-provoking, and it opens up several avenues for future inquiry into how scientific writing style intersects with identity and recognition. With some additional attention to interpretive framing and theoretical nuance—particularly around gender and impact—I believe this paper will make a valuable contribution to ongoing conversations around science communication and structural inequality in academia.

Version 1:

Decision Letter:

Dear Professor Stavrova,

Your manuscript titled "Scientific publications that use promotional language receive more citations and public attention" has now been seen by our reviewers, whose comments appear below.

Reviewers #1 and #3 are referees from the previous round, Reviewer #4 is an ECR co-reviewer who previously submitted a joint report with Reviewer #3.

In light of their advice I am delighted to say that we are happy, in principle, to publish a suitably revised version in Communications Psychology.

We therefore invite you to revise your paper one last time to address a list of editorial requests. At the same time we ask that you edit your manuscript to comply with our format requirements and to maximise the accessibility and therefore the impact of your work.

EDITORIAL REQUESTS:

I highlight in particular that in-text statistics reporting should be more comprehensive, and that the Figures will require revisions to align with our requirements for data presentation.

SUBMISSION INFORMATION:

OPEN ACCESS:

* DATA AVAILABILITY:

Link Redacted

Best regards,

Marika Schiffer, on behalf of

Daniel Quintana

Daniel Quintana, PhD
Editorial Board Member
Communications Psychology
orcid.org/0000-0003-2876-0004

REVIEWERS' COMMENTS:

Reviewer #1 (Remarks to the Author):

I appreciate the author's revisions. All of my previous comments have been addressed, and I have no additional feedback at this time.

Reviewer #3 (Remarks to the Author):

The authors did a thorough revision, addressing all of our comments.

Reviewer #4 (Remarks to the Author):

This paper offers a compelling and large-scale analysis of promotional language in scientific abstracts and its relationship to both academic and public impact. The central claim is that promotional language is associated with greater visibility and citations, and that these benefits are unevenly distributed by gender. This is important, timely, and likely to spark meaningful conversation across fields.

The scope of the dataset is impressive, and the statistical modeling is both rigorous and transparent! I also appreciated the care with which gender differences were analyzed... as a theoretically motivated question about inequality in recognition.

That said, a few areas left me wanting more clarity or depth:

On the meaning of "impact": The paper treats citation counts and online mentions as straightforwardly desirable. But given the gendered differences you document, and the known biases in citation and media attention, I wondered if a brief reflection on the limits of "impact" as currently measured might help contextualize the findings. Is promotional language increasing visibility in ways that reward attention-seeking more than substance?

Distinction between promotional and positive language: You do explain that your dictionary captures more than just "positivity," but I think this could be clarified earlier in the paper. A few examples of promotional-but-not-positive terms in the introduction (not just the supplement) would help readers understand what's being measured.

Field and disciplinary variation: While you do control for field in the models, I wondered whether some disciplines are more likely to be flagged as promotional simply due to jargon (e.g., "novel" in medicine vs. "stellar" in astrophysics). A short comment on this possibility—perhaps in the discussion—might be worth including.

The gender framing: The results about widening gender gaps with promotional language are fascinating. I appreciated your revised introduction softening the claim that gender is the key moderator. That change makes the narrative feel more measured. One thing I was curious about—but may be out of scope—is whether the backlash effect might look different in single-author vs. multi-author papers. Just a thought for future work.

Overall, this is a well-designed and thought-provoking manuscript. It's rare to see a study that's both statistically ambitious and socially meaningful. With just a few clarifications and some added nuance in framing, I believe this paper will be a strong contribution.

Comment 1:

The paper examines how promotional language in scientific publications influences academic impact and public engagement, using a dataset of over 130,000 abstracts from Science, Nature, and PNAS (1991–2023). It finds that promotional language is associated with higher citation rates, increased full-length views, and greater Altmetric scores, as well as enhanced public engagement through social media and blog mentions. The study also highlights gender disparities, showing that female scientists use promotional language less frequently and do not benefit from it as much as their male counterparts.

Strengths:

- + Large dataset covering three decades and high-impact journals.
- + Multiple impact metrics (citations, full-text views, Altmetric scores).
- + Robust statistical methods with different statistical measures.

Weaknesses:

- Correlation does not imply causation. While the study shows a link between promotional language and impact, it does not establish causality. Factors like research quality may influence both promotional language use and citation counts. Experimental testing could strengthen conclusions.

Response: we fully agree with this view and revised the manuscript to remove any causal language. We also note the possibility of alternative causal explanations (e.g., research quality) in the discussion (p. 26, second paragraph).

Comment 2:

- Papers with stronger findings may naturally use more positive language, while those with less significant or negative results may tone down claims. It remains unclear whether promotional language itself drives impact or if it simply reflects the nature of the results.

Response:

We discuss the possibility of the use of promotional language reflecting the nature of results, including their significance, on p. 26 (second paragraph). Further, the research papers we analyzed were published in three top interdisciplinary journals, which is likely to minimize the between-paper differences in whether they present significant (both statistically and conceptually) results or not. We also would like to note that we do use “positive language” (what we refer to as “semantic positivity”, i.e. the share of positive-valence words) as a control variable and find that the effect of promotional language persists when controlling for positive language (see Table 1). Also, it is interesting to note that promotional and positive language are sufficiently distinct and correlate only weakly : $r = .14, p < .001$. Promotional language includes negative terms as well: *alarming, elusive, daunting, dismal, devastating, unanswered, unmet*, to give a few example (the list of all words in the promotional language dictionary is presented in the Supplementary Information File), while positive language does not.

Comment 3:

- The study uses automated name-based gender classification, which can misclassify non-binary individuals and certain cultural names. A more refined classification method would improve accuracy.

Response: We added a validation analysis that showed very high convergence between automatic and manual name detection (added on p. 9). We further discuss the weaknesses of the gender classification based on first names in the discussion (starting on p. 25, last paragraph).

Comment 4:

-The study focuses on only three high-impact journals (Science, Nature, PNAS). This excludes disciplinary differences and journals with different editorial styles. Expanding to a wider range of journals would improve generalizability.

Response: The journals we included are interdisciplinary, covering a broad range of disciplines, from mathematics, to biology, to health sciences. We added a comment about extending the analyses for a broader scope of journals to improve generalizability in the discussion (see p. 27, starting on line 3).

Comment 5:

While this paper contributes to ongoing discussions on science communication and linguistic framing, similar research exists.

1. Edlinger et al. (2023)

Found that positive language in abstracts has increased and is strategically placed to influence perception. Given this, how does the current study add to the understanding beyond showing citation benefits?

Edlinger, M., Buchrieser, F., & Wood, G. (2023). Presence and consequences of positive words in scientific abstracts. *Scientometrics*, 128(12), 6633-6657.

Response: This paper focuses on the use of positive language (not promotional language – as our paper; see our response to your comment 2) in abstracts and shows that such use has increased in the last decades. It does not test whether the use of positive language predicts any indicators of impact. In contrast, our paper shows that the use of promotional language (irrespective of positive language) is associated with more impact (in terms of citation count and public attention) and that this effect depends on authors' gender.

2. Lerchenmueller et al. (BMJ 2019)

Found that positive framing in scientific abstracts correlates with higher citation counts and that male scientists use promotional language more often. How does this study advance beyond confirming these findings?

Lerchenmueller, M. J., Sorenson, O., & Jena, A. B. (2019). Gender differences in how scientists present the importance of their research: observational study. *BMJ*, 367.

Response: thank you for directing our attention at this paper. As the Edlinger et al.'s paper, this paper focuses on positive (not promotional) language and tests whether its use depends on the combination of the genders of the first and last author. Instead, our research focuses on promotional language (which is related but not equivalent to positive language – see our response to your comment 2).

Besides this important difference between the work of Lerchenmueller et al. and ours, Lerchenmueller et al. show that the use of positive words in abstracts is associated with higher citations numbers of these papers but do not test whether this equally applies to papers with female vs. male authors. In sum, this paper is highly relevant for our research and we now cite it on p. 5 (starting line 4). However, our research is sufficiently distinct as it 1) studies promotional language (while controlling for positive language), 2) assesses its effect not only on citation count but also on broader indicators of impact and public interest, and 3) tests whether the use of promotional

language predicts citation count and other indicators of impact differently for male vs. female authors.

3. Wu et al. (2024)

Found that positive words predict higher citation counts but noted a declining citation advantage over time. In contrast, this study suggests that promotional language continues to boost citations and engagement. Could the authors explain this discrepancy?

Wu, D., Wu, H., & Li, J. (2024). Citation advantage of positive words: predictability, temporal evolution, and universality in varied quality journals. *Scientometrics*, 129(7), 4275-4293.

Response:

Our original analyses included publication year just as a control variable. In the revision, to test whether the association of promotional language with impact indicators diminishes with time, we conducted additional analyses with impact indicators as dependent variables, promotional language, publication year and the interaction between the two as predictors. The interaction term was significant with respect to all impact indicators and persisted when including the control variables (the same as used in the main analyses, see the paragraph "Control variables" in the Methods section). The pattern of the interaction depended on the specific impact indicator used. Specifically, we found that the association between promotional language and impact has diminished over time when impact was measured as citation count per year. In contrast, with respect to other five indicators of impact (number of views per year and in the last 180 days, Altmetric score, number of mentions on Twitter and overall number of mentions in online posts), the positive association between promotional language use and impact increased with time. We would like to re-emphasize that promotional and positive language are not the same and hence our results cannot be directly compared to Wu et al. Nevertheless, we now added this dynamic temporal perspective to the results (p. 13) and reflect on it in the discussion (p. 27, third paragraph).

Reviewer #2 (Remarks to the Author):

Comment 1:

This study examines the potential effects of promotional language in scientific publications published in three iconic venues (Science, Nature and PNAS). The authors rely on specific terms in the publications' abstracts to do this.

Overall, the work presents objectives of great interest to the meta-research community and quantitative studies in science. The manuscript is very well structured and written. The methodology followed especially the statistical component, is robust and provides highly interesting results for understanding the effect of the use of populist terms on the impact of the papers. The conclusions are also derived from the data and are nuanced and contextualized. For this reason, I am positive about the publication of this work.

Response: thank you for the overall positive evaluation of our work.

Comment 2:

However, I note some issues that I will address below in order to improve the justification of some decisions and enhance the manuscript.

Introduction

On line 131, the authors indicate "across over 50,000 publications." However, I cannot see where this number of publications is, as 130,000 abstracts were analyzed after the filtering process.

Response: indeed, the analyses of the effect of promotional language on impact are based on 130,000 abstracts but gender-related analyses are based on ~50,000 abstracts, because we only included the abstracts for which the first and last author gender could be unequivocally identified. For some of the records, the complete first names were not available (e.g., M. Smith) and for some other records, the first names were unisex. We clarify this in the revised manuscript on p. 9, first paragraph, and discuss this in the limitations section of the discussion (p. 25, last paragraph).

Comment 3:

As regards the gender issue, the authors introduce the topic indicating that "There is one factor that is particularly likely to affect the effectiveness of promotional language: author gender". However, from my point of view, the gender issue should be introduced in the study in a slightly different way, specifically as one, among many others, of the potential consequences of promotional language. The question is whether promotional language is more effective according to the author's gender, considering that women tend, on average, to use less promotional language. Consequently, the promotional language should be an additional barrier to gaining citations and visibility.

Response: We have revised how we introduce the gender topic following your recommendation (see p. 5).

Comment 4:

Method

2023 is a very recent publication to consider the number of citations, especially considering that data was collected by June 2023. The authors indicate this issue later on lines 209 and 210, mentioning the altmetrics collected. Publications before 2011/2012 cannot be compared to other recent publications.

Response:

We conducted additional robustness checks while excluding publications that appeared in 2023. The results were consistent with the ones reported in the main text and are shown in Supplementary Tables 7-9.

Also, please note that the Altmetric score is available for publications that appeared before Altmetric started collecting data. This is because even older papers receive public attention and are mentioned on Twitter/X and other online resources. However, we agree that the public attention to more recent papers might be higher, therefore one of our robustness check analyses excluded papers published before 2010. The results of these analyses were also consistent with the main findings (see Supplementary Table 1-3).

Comment 5:

Otherwise, the authors use the Altmetric attention score. However, this indicator has been described in the literature as an opaque and non-responsible metric not recommended for bibliometric studies. Authors should justify using this composed indicator better than raw metrics. In this sense, the number of tweets is one of the most important altmetric indicators. Still, the number of Mendeley readers could have provided other interesting findings and the presence of publications in news and press releases.

Response: We acknowledge the lack of transparency in how the Altmetric score is calculated in the discussion (p. 24, starting last paragraph). That is also the reason why we complemented the Altmetric score analysis with raw data of mentions on Twitter/X and in posts. Altmetric API does not provide the same indicators for all the records. E.g., for some papers, the number of mentions in patents and policy documents is available, while for others, it is the number of mentions on Wikipedia. We selected the raw count of mentions on Twitter, as Twitter has been one of the most popular online social networks for science communication at the time of writing. We selected the raw number of posts as this metric reflects the number of all mentions in all types of online documents tracked by Altmetric (Facebook, LinkedIn, blogs, reddit, news etc.: <https://details-page-api-docs.altmetric.com/data-endpoints-fetch.html#posts>). We apologize for confusing the reader by occasionally referring to this metric as “blogposts”, we adjusted our language in the revision and provided more details regarding the origin and the exact composition of the metrics we use (see p. 8, second paragraph, and in the note to the tables). Further, in our experience back during the time of data collection, the number of mentions in tweets and posts were the two raw metrics most consistently available for the largest majority of the records. We followed a similar logic with regard to the measure of the number of readers. Here, we relied on the number provided by WoS (number of record views on the WoS platform) as it was available for all records in our dataset. In contrast, obtaining the number of Mendeley readers would have resulted in data loss as it was only available via the Altmetric API which did not work for all the records in our dataset (see p. 8, second paragraph, in the manuscript). For example, some of the records from WoS were just not listed in the Altmetric API (example: <https://api.altmetric.com/v1/doi/10.1126/science.1010580>).

Comment 6:

While the authors use several metrics to control de variability, I suggest that future studies collect field-normalized citation-based data.

Response: we agree and discuss this possibility on p. 24 (second paragraph).

Comment 7:

Results

I probably misunderstood the logic, but the authors say, “last author gender was not significantly associated with the share of promotional words” (lines 242/243). However, the authors say later, “Again, the impact gap between female- and male-last author papers increased with increasing promotional language” (lines 256/257). Later, in Figure 2 panel B, we can observe significant differences for the last author in all control metrics, including sharing promotional words. It was somewhat confusing to follow the authors’ argument.

Response: the sentence “last author gender was not significantly associated with the share of promotional words” refers to a gender difference in the use of promotional words (Table 2). To avoid confusion, we rephrased it as “Abstracts of papers with female (vs. male) first author included 6% less promotional language; in contrast, abstracts of papers with a female last author did not differ in the share of promotional words compared to the abstracts of papers with a male last author”. The sentence “Again, the impact gap between female- and male-last author papers increased with increasing promotional language” describes the last author gender x promotional language interaction effect when predicting impact. It says that the gender gap in citations and other indicators of impact is larger for papers that use more (vs. less) promotional language. We hope that this provides the necessary clarification.

Comment 8:

Discussion

As the authors already indicate, causal conclusions cannot be driven. In addition, the quality or novelty of papers should not be discussed here. No assumption should be made about using promotional language to indicate that the study is less serious, especially when analyzing only the abstract of publications. Abstracts can be written to catch the readers (sometimes even requested by the journals), but the full-text manuscript can be accurate.

Moreover, the study covered three special multidisciplinary journals, with huge coverage and strong communication activities, and was frequently mentioned in the press. I don't think that these journals can represent the scientific activity. These aren't "normal journals", bibliometrically speaking. A comment on this issue is necessary, in my opinion.

Response: we made sure to remove the assumption that promotional language might be a signal of less serious work and we added a consideration regarding how our results could generalize to a broader range of journals (p. 27, second paragraph).

Comment 9:

Otherwise, the authors do not mention other studies focused on using promotional language in the titles that can add background to this manuscript. In this same line, I miss a discussion on the effect of AI tools when writing publications. Literature has already evidenced that specific terms are overused due to AI tools that can affect visibility and readability.

Response: we included more citations of studies that explore promotional language. However, please note that we do not cover all the literature on the use of positive language, as we consider it sufficiently distinct from promotional language (see our response to Comment 2 of Reviewer 1). We really liked the idea of how the advancement of generative AI might shape the academic language use and added a discussion on the role of AI tools on p.

Comment 10:

Finally, future studies should use credit taxonomy to analyze the effects of promotional language by author gender, considering the person with the credit for writing the manuscript, regardless of their position.

Response: We now discuss this point on p. 27 (last paragraph).

Comment 11:

Minor issues:

Please, fix some typos: "assertive language" "reent study" "scientsts"

Keywords should be improved, as relevant terms are omitted, such as gender analysis or scientific promotion.

References 21 and 22 seem the same. Please check.

Response: thank you, we corrected these minor issues.

Reviewer #3 (Remarks to the Author):

Comment 1:

This manuscript offers a large-scale and methodologically sound analysis of over 130,000 scientific abstracts, examining whether the use of promotional language is associated with greater academic and public impact. The findings are compelling: promotional language is positively associated with citation counts and media visibility, but the effects are not equally distributed across gender lines. Specifically, promotional language tends to amplify the existing gender citation and visibility gap rather than mitigate it, with male authors benefiting more than their female counterparts. This work is important and timely. The authors tackle a subject at the intersection of metascience, communication psychology, and gender studies, offering empirical clarity on a cultural practice that has long simmered under anecdotal and editorial scrutiny. We commend the authors for their ambitious scope and analytical rigor. That said, we would like to see more interpretive clarity and a deeper discussion, particularly around gender inference, the normativity of “impact,” and the operationalization of promotional language. These areas would benefit from further reflection.

Response: thank you highlighting the strengths of our work, we really appreciate it.

Comment 2:

Major comments:

1. The meaning and normativity of “impact”

The authors present impact as an inherently desirable outcome (e.g., more citations, higher Altmetric scores). We understand that citations are the de facto currency in academia, but would encourage the authors to reflect more critically on what these metrics actually signify. To what extent do citations or tweet counts indicate scientific quality or social value, and to what extent do they simply reflect success within the attention economy, shaped by performative cues and systemic inequities? This question becomes especially salient in light of the gendered dynamics at play: if men are rewarded more for using promotional language, are we witnessing meritocratic advantage, or a socially-coded responsiveness to assertive (often male-coded) behaviors? It might strengthen the discussion to include a brief reflection on the value-neutrality (or lack thereof) of “impact” metrics, and what that means for interpreting the findings.

Response: thank you for this suggestion. We agree with this sentiment and have included these points into the discussion (p. 24, second paragraph).

Comment 3:

2. Operationalization of promotional language

The dictionary-based approach is reasonable and well-situated within prior literature. However, we are wondering whether the paper engages deeply enough with the issue of polysemy and disciplinary norms. Words like novel or robust may be promotional in one context but simply technical in another. Could some disciplines (e.g., astrophysics vs. medicine) be systematically flagged as “more promotional” due to these differences? We'd be curious to hear whether the authors tested for dictionary validity across time or field, and whether certain domains were disproportionately flagged for high promotional word use.

Response: thank you for this reflection. Please note that our control variables included the academic field (discipline) and year of publication (also, in newly conducted additional analyses we tested whether the effect of promotional language on impact changes over time) and our results were robust against these control variables. Further, we think that one of our robustness checks speaks to your concern more directly: as some of the “promotional language” terms might have several meanings (e.g., “stellar” as “exceptional” vs. describing stars in astronomy) and can thus artificially inflate the promotional language score for some papers, we repeated the analyses while excluding

abstracts with an exceptionally high (3 SD above the mean) share of promotional words. The results were consistent with the main analysis and are reported in Supplementary Tables 4-6). Further, we added a discussion on the limitations of the dictionary approach on p. 25 (second paragraph).

Comment 4:

3. Gender inference and cultural bias

The authors rightly note the limitations of using name-based gender classification, particularly for Asian names. However, the claim that this ambiguity may “protect” some authors from gender bias feels speculative and potentially problematic. Framing it that way risks naturalizing a Western-centric lens on name recognition and gender signaling. Rather than describing gender ambiguity as a protective shield, perhaps frame it as a gap in methodological inclusivity, one that future work could address through cross-referencing institutional data or author self-identification. Additionally, we’d encourage the authors to acknowledge that race, region, institutional prestige, and other intersecting identities may also shape both promotional strategies and their downstream effects. These dynamics deserve more attention, even if they are beyond the scope of the current dataset.

Response: we followed your advice in reframing and discussing this issue (p. 25, last paragraph).

Comment 5:

4. The backlash hypothesis

The authors gesture toward the backlash hypothesis but also acknowledge that the findings only partially support it. We appreciated this nuance, though at times the conclusion seems to stretch slightly beyond what the data warrant. Observing smaller gains for women is not the same as showing that self-promotion hurts women. But it certainly invites questions about why women may (or may not) engage in self-promotion at the same rate as men. This could be a good opportunity to incorporate perspectives from impression formation or gendered communication research, which might offer alternative explanations beyond backlash, such as strategic ambiguity, likability penalties, or status expectation mismatches.

Response: We adjusted our phrasing to make sure that we refer to the absence of a negative effect of promotional language for women: “While the link between promotional language and higher impact may be smaller for females than for males, we found no evidence of more (vs. less) promotional language being linked to a lower impact in women.” (see p. 23, first paragraph). We further followed your suggestion and included a reflection on why women use promotional language less than men on p. 23.

Comment 6:

Minor comments:

5. Does all promotional language count as “hyping?” The paper could do more than providing a few example keywords to train the reader's intuition about what kind of self-promotional language is studied.

Response: we provided a definition of promotional language and offered more examples on p. 3 (last line).

Comment 7:

6. The manuscript comments on possible issues with the relevancy of views (i.e., the number of times the paper's record has been accessed on the Web of Science platform). It seems that the two analyses based on views are robustness checks, complementing the investigation focused on citations.

Response: All metrics have their limitations and they do partially lead to different findings (in particular, see our new exploratory analyses of temporal dynamics – p. 13). Therefore, also aligned with the feedback of other reviewers, we decided to report all the outcome measures in the manuscript.

Comment 8:

7. When the authors are quantifying citations, they use the yearly average citations even though we know that citations are not accumulated linearly. The paper could acknowledge this possible shortcoming in measurement.

Response: We acknowledge that even though we calculate citation count per year, the year might still affect citations. Therefore, we control for the publication year in the analyses. In the revision, we also provide additional analyses showing that the association of promotional language with impact changes over time. We present (p. 13) and discuss these new findings (p. 27, paragraph 3) in the revision.

Comment 9:

8. The Altmetric score includes mentions in blogposts and on Twitter. As the paper acknowledges, these three indicators are somewhat correlated, so here too it may be worth showing one of them and use the others as robustness checks. The current title could guide the selection of what may constitute the main analysis.

Response: Please see our response to comment 5 of Reviewer 2. Also, we decided to indeed adjust the title a little to make it more inclusive of the outcome variables: "Scientific publications that use promotional language receive more citations and public attention".

Comment 10:

9. Please mention the rationale behind recoding of the 20 fields and list which fields belong to the three aggregated groups.

Response: Some of the 20 fields included a very small number of papers: e.g., there was only 1 paper in the field of philosophy and religious sciences, 14 in the field of legal studies and 15 in the field of creative arts and writing sciences (to compare, the most well-represented field – biological sciences – included almost 65,000 records). We included this information p. 10 (first paragraph). We added the list of the fields within each of the three broader category in Supplementary Information File.

Comment 11:

10. Can you please specify the fit of the models in Table 1?

Response: For negative binomial models, traditional fit indices like RMSEA or CFI (used in SEM) are not typically available. We could report AIC and BIC information criteria but they cannot be interpreted as indicators of model fit. We included Nagelkerke's R^2 instead (for negative binomial models and multiple and adjusted R^2 for linear models, see Tables 1, 3, 4).

Comment 12:

11. It seems prudent to stay away from suggestions like "testing whether authors can boost the impact of lower-quality work simply by using more promotional language."

Response: we rephrased this sentence: "We nevertheless strongly encourage future work to consider experimental study designs that would allow to draw causal conclusion or using alternative indicators

of work quality (e.g., evaluations during peer review), testing whether authors can boost their work's impact by simply using more promotional language.”

Comment 13:

This is a compelling and necessary paper. It offers strong empirical evidence for an effect that many have intuited but few have rigorously demonstrated: that promotional language boosts visibility—but not equitably. We found the gender moderation analysis particularly thought-provoking, and it opens up several avenues for future inquiry into how scientific writing style intersects with identity and recognition. With some additional attention to interpretive framing and theoretical nuance—particularly around gender and impact—I believe this paper will make a valuable contribution to ongoing conversations around science communication and structural inequality in academia.

Response: thank you a lot for highlighting the strengths of our work. We are confident that your suggestions have further improved its quality.

Reviewer #4 (Remarks to the Author):

This paper offers a compelling and large-scale analysis of promotional language in scientific abstracts and its relationship to both academic and public impact. The central claim is that promotional language is associated with greater visibility and citations, and that these benefits are unevenly distributed by gender. This is important, timely, and likely to spark meaningful conversation across fields.

The scope of the dataset is impressive, and the statistical modeling is both rigorous and transparent! I also appreciated the care with which gender differences were analyzed... as a theoretically motivated question about inequality in recognition.

That said, a few areas left me wanting more clarity or depth:

On the meaning of “impact”: The paper treats citation counts and online mentions as straightforwardly desirable. But given the gendered differences you document, and the known biases in citation and media attention, I wondered if a brief reflection on the limits of “impact” as currently measured might help contextualize the findings. Is promotional language increasing visibility in ways that reward attention-seeking more than substance?

Response: We added this reflection on p. 29, second paragraph.

Distinction between promotional and positive language: You do explain that your dictionary captures more than just “positivity,” but I think this could be clarified earlier in the paper. A few examples of promotional-but-not-positive terms in the introduction (not just the supplement) would help readers understand what’s being measured.

Response: We added this clarification earlier in the introduction (see p. 4., first line).

Field and disciplinary variation: While you do control for field in the models, I wondered whether some disciplines are more likely to be flagged as promotional simply due to jargon (e.g., “novel” in medicine vs. “stellar” in astrophysics). A short comment on this possibility—perhaps in the discussion—might be worth including.

Response: We added this information on p. 29, first paragraph.

The gender framing: The results about widening gender gaps with promotional language are fascinating. I appreciated your revised introduction softening the claim that gender is the key moderator. That change makes the narrative feel more measured. One thing I was curious about—but may be out of scope—is whether the backlash effect might look different in single-author vs. multi-author papers. Just a thought for future work.

Response: thank you for these thoughts. Indeed, the number of authors might be a relevant moderator here. We included this reflection in the discussion on p. 28, first paragraph.

Overall, this is a well-designed and thought-provoking manuscript. It’s rare to see a study that’s both statistically ambitious and socially meaningful. With just a few clarifications and some added nuance in framing, I believe this paper will be a strong contribution.

Response: we thank you for a positive evaluation of our work and the thoughtful improvement suggestions.